

# From widespread Mississippian to localized Pennsylvanian extension in central Spitsbergen, Svalbard

**Jean-Baptiste P. Koehl[1,2], Jhon M. Munoz-Barrera[3]**

[1]Department of Geosciences, UiT The Arctic University of Norway in Tromsø, 9037 Tromsø, Norway.

[2]Research Centre for Arctic Petroleum Exploration (ARCEx), UiT The Arctic University of Norway in Tromsø, 9037 Tromsø, Norway.

[3]Department of Earth Science, University of Bergen, Postboks 7803, 5020 Bergen, Norway.

**Correspondence:** Jean-Baptiste P. Koehl (jean-baptiste.koehl@uit.no)

## Abstract

In the Devonian–Carboniferous, a rapid succession of clustered extensional and contractional tectonic events is thought to have affected sedimentary rocks in central Spitsbergen. These events include Caledonian post-orogenic extensional collapse associated with the formation of thick Early–Middle Devonian basins, Late Devonian–Mississippian Ellesmerian contraction, and Early–Middle Pennsylvanian rifting, which resulted in the deposition of thick sedimentary units in Carboniferous basins like the Billefjorden Trough. The clustering of these varied tectonic settings makes it sometimes difficult to resolve the tectono-sedimentary history of individual stratigraphic units. Notably, the context of deposition of Mississippian clastic and coal-bearing sedimentary rocks of the Billefjorden Group is still debated, especially in central Spitsbergen. We present field evidence from the northern part of the Billefjorden Trough, in Odellfjellet (Austfjorden), suggesting that tilted Mississippian sedimentary strata of the Billefjorden Group deposited during active (Late/latest?) Mississippian extension. Evidence include slickenside lineations and growth strata in the hanging wall of basin-oblique NNE-dipping faults, such as the Overgangshytta fault. These basin-oblique faults systematically die out upwards within Mississippian to lowermost Pennsylvanian strata and suggest a period of widespread WNW–ESE-directed extension in the Mississippian (rift "initiation" phase), followed by an episode of more localized extension in Early–Middle Pennsylvanian times ("interaction and linkage" and "through-going fault" phases). In addition, the presence of abundant basin-oblique faults parallel to the





Overgangshytta fault in basement rocks adjacent to the Billefjorden Trough suggests that the formation of Mississippian normal faults was partly controlled by reactivation of preexisting Neoproterozoic (Timanian?) basement-seated fault zones. We propose that these existing faults reactivated as transverse fault or accommodation cross faults in or near the crest of transverse folds reflecting differential displacement along the Billefjorden Fault Zone, thus suggesting that normal

faulting along this major fault initiated as early as the Mississippian. In Cenozoic times, the Overgangshytta fault may have mildly reactivated as an oblique thrust during transpression– contraction, and shallow-dipping bedding-parallel duplex-shaped decollements in shales of the Billefjorden Group possibly prevented further movement along Mississippian margin-oblique faults.

## 1. Introduction

At the end of the Caledonian Orogeny in late Paleozoic times, Norway (Séranne et al., 1989; Osmundsen and Andersen, 2001; Gudlaugsson et al., 1998; Koehl et al., 2018a), Greenland (Hartz et al., 1997; Sartini-Rideout et al., 2006; Hallett et al., 2014; McClelland et al., 2016) and Svalbard

(Manby and Lyberis, 1992; Braathen et al., 2018) were part of a large E–W trending intra-cratonic basin (Ziegler et al., 2002) that was subjected to a major episode of gravitational collapse, resulting in the formation of thick, Early to Middle Devonian sedimentary basins that evolved into rift basins in Late Devonian (?) to Carboniferous times. In Spitsbergen, however, Late Devonian– Mississippian times recorded a short-lived period of contraction related to the Ellesmerian

Orogeny, inverting Devonian collapse basins and associated basin-bounding faults (Piepjohn, 2000; Bergh et al., 2011; Piepjohn et al., 2015). Further transpression related to the opening of the Northeast Atlantic Ocean and the formation of a major fold-and-thrust belt in Cenozoic times complicates the study of Mississippian sedimentary rocks, making it difficult to identify and resolve Mississippian fault movements. Although the sedimentology and stratigraphy of

Mississippian sedimentary rocks are well studied in Spitsbergen (Gjelberg and Steel, 1981; Gjelberg, 1984; McCann and Dallmann, 1996; Maher, 1996), Bjørnøya (Gjelberg, 1981; Gjelberg and Steel, 1983; Worsley et al., 2001) and the SW Barents Sea (Bugge et al., 1995; Larssen et al., 2002; Samuelsberg et al., 2003; Koehl et al., 2018a), little is known about the tectonic setting in which they were deposited. Particularly in central Spitsbergen, in the Billefjorden Trough

(Braathen et al., 2011), Mississippian sedimentary rocks are believed to represent pre-rift



sedimentary rocks deposited prior to the main phase of extension in the Pennsylvanian (Johannessen and Steel, 1992; Braathen et al., 2011). However, new field observations in Mississippian strata in Austfjorden, in the northern part of the Billefjorden Trough (Figure 1), challenge this model.

The present study provides new insights in the Mississippian tectonic history of central Spitsbergen, Svalbard, using field structural analysis of newly exposed Mississippian sedimentary deposits in Odellfjellet, Austfjorden (Figure 1). These sedimentary rocks are mildly reworked by Cenozoic transpression, and show preserved Mississippian primary faults and offsets, thus representing an excellent opportunity to resolve the tectonic history of this period. We emphasize

the influence of NW–SE-striking faults, like the Overgangshytta fault, on the deposition of Mississippian–Lower Pennsylvanian sedimentary strata and use adjacent and/or overlying Lower– Late Pennsylvanian sedimentary rocks as a comparison. Finally, we discuss potential controlling factors that may have influenced Mississippian faulting.

**2.  Geological setting**

**Precambrian geology**

The study area, the Billefjorden Trough, is located at the boundary of two major structural domains, the northwestern and eastern terranes of Svalbard (Harland and Wright, 1979; Ohta et al., 1989; Labrousse et al., 2008), previously named the Nordfjorden and Ny-Friesland blocks

respectively (Cutbill and Challinor, 1965; Harland et al., 1974). East of the trough, the Ny-Friesland block is composed of basement rocks with well developed, variably dipping, N–S-trending foliation, dominated by biotite-amphibolite gneisses of the Eskolabreen Complex (Balashov et al., 1993; Johansson and Gee, 1999) and Meso- to Neo-proterozoic metasedimentary rocks of the Smutsbreen and Polhem formations (Harland et al., 1966). These rocks are involved

in a large-scale, N–S-trending, gently north-plunging fold structure, the Atomfjella Antiform (Witt-Nilsson et al., 1998). In addition, Paleoproterozoic granitic and granodioritic basement gneisses (Harland et al., 1974) crop out in the hanging wall of the Balliolbreen fault and in the footwall of the Odellfjellet fault, two major segments of a regional east- to ENE-dipping fault complex, the Billefjorden Fault Zone (BFZ; Harland et al., 1974; McCann and Dallmann, 1996; Braathen et al.,

2011; Figure 1).





**Late Paleozoic post-Caledonian basins and faults**

*Devonian sedimentary basins*

Post-Caledonian "Old Red" collapse basins formed along inverted Caledonian thrusts in

the Early to Late Devonian and are bounded by major N–S- to NNW–SSE-striking faults (Harland et al., 1974; Manby and Lyberis, 1992; Manby et al., 1994). Large portions (> 6 km-thick) of these basins are preserved west of a west-dipping segment of the BFZ, although they were probably deposited east of the fault as well (McCann and Dallmann, 1996). Devonian collapse sediments were possibly reworked by contraction related to the Late Devonian–Mississippian Svalbardian

Phase (McCann, 2000; Piepjohn, 2000; Bergh et al., 2011; Piepjohn et al., 2015). Notably, in Billefjorden and Austfjorden (Figure 1), positive tectonic inversion of the Balliolbreen segment of the BFZ resulted in over-thrusting and juxtaposition of Paleoproterozoic, granitic and granodioritic basement gneisses to the east with Devonian clastic sedimentary deposits to the west (Figure 1; McCann, 2000). However, this short-lived episode of contraction is challenged by new evidence

of basement exhumation, possibly as core complexes along inverted Caledonian shear zones in Early to Late Devonian times in northwestern Spitsbergen (Braathen et al., 2018), in Early Devonian to Mississippian times in the SW Barents Sea (Klein and Steltenpohl, 1999; Klein et al., 1999; Steltenpohl et al., 2011; Koehl et al., 2018a), and in the Late Devonian–Mississippian in northeastern Greenland (Sartini-Rideout et al., 2006; Hallett et al., 2014; McClelland et al., 2016).


*Carboniferous sedimentary basins*

During post-Caledonian, Carboniferous, ENE–WSW-directed extension/sinistral transtension, multiple sedimentary troughs formed throughout the Svalbard archipelago, e.g., the Billefjorden, Lomfjorden, St Jonsfjorden and Inner Hornsund troughs (Maher, 1996; McCann and

Dallmann, 1996), while major sedimentary basins, such as the Sørkapp, Nordkapp and Hammerfest basins, developed in the Barents Sea (Gabrielsen et al., 1990; Gudlaugsson et al., 1998; Anell et al., 2016; Koehl et al., 2018a). These basins and troughs were filled with thick Carboniferous sediments deposited along (reactivated) high-angle normal faults, like the east-dipping Balliolbreen and Odellfjellet segments of the BFZ in central Spitsbergen (Harland et al., 1974; McCann and

Dallmann, 1996).

Mississippian sedimentary strata are up to 2.5 km in cumulative thickness, and are easily recognizable at outcrop scale because they commonly comprise coal seams and coaly shales





interbedded with dominant clastic deposits, both in the Barents Sea (Bugge et al., 1995; Larssen et al., 2002; Samuelsberg et al., 2003), on Bjørnøya (Gjelberg and Steel, 1983; Gjelberg, 1984) and

in Spitsbergen (Cutbill and Challinor, 1965; Cutbill et al., 1976; Gjelberg, 1981, 1984; Gjelberg and Steel, 1981). In central Spitsbergen (e.g., in Billefjorden), preserved Mississippian strata are relatively thin (< 300 m; Cutbill et al., 1976) and are divided into two formations, the Hørbyebreen Formation composed of the Triungen and Hoelbreen members, and the Mumien Formation including the Sporehøgda and Birger Johnsonfjellet members (Figure 2). The Hoelbreen and Birger

Johnsonfjellet members show abundant, characteristic coal seams and coaly shales, whereas the Triungen and Sporehøgda members are dominantly composed of clastic sedimentary deposits (Cutbill and Challinor, 1965; Cutbill et al., 1976; Gjelberg and Steel, 1981; Gjelberg, 1984; Figure 2).

Mississippian sedimentary rocks of the Billefjorden Group are generally believed to represent pre-rift units (Johannessen and Steel, 1992; Braathen et al., 2011), though an early syn-

rift origin is considered possible (Steel and Worsley, 1984; Nøttvedt et al., 1993; McCann and Dallmann, 1996). The pre-rift interpretation is largely based on the presence of Mississippian rocks on both sides of the BFZ. Moreover, Mississippian sedimentary strata display NW-plunging folds (e.g., in western Spitsbergen), suggesting that they might have (partly) deposited during contraction

related to the Svalbardian phase (Bergh et al., 2011) of the Late Devonian–Mississippian Ellesmerian Orogeny (McCann, 2000; Piepjohn, 2000). During this contractional event, the BFZ might have acted as a transpressional fault, possibly accommodating left-lateral displacement > 200 km (Harland et al., 1974). In addition, contraction-related uplift may be responsible for extensive erosion of Mississippian rocks, thus, preventing direct comparison of sedimentary

successions in faults footwall and hanging wall and impeding the identification of potential growth strata (McCann and Dallmann, 1996).

Pennsylvanian sedimentary rocks in central Spitsbergen represent the thickest, preserved sedimentary deposits recorded in the Billefjorden Trough. These are divided into five formations belonging to the Gipsdalen Group (Figure 2). First, the late Serpukhovian Hultberget Formation is

composed of characteristic red and subsidiary grey sandstones, conglomerates and shales (Cutbill and Challinor, 1965; Cutbill et al., 1976; Johannessen, 1980; Gjelberg and Steel, 1981; Johannessen and Steel, 1992; Figure 2). Second, the Bashkirian Ebbadalen Formation is made of highly variable lithologies, including interbedded grey–yellow sandstones and grey–green shales (Ebbaelva



Member), and red and yellow sandstones and conglomerates interbedded with red shales
(Odellfjellet Member) interfingering with gypsum–anhydrite and dark limestones and dolomites
(Trikolorfjellet Member; Holliday and Cutbill, 1972; Johannessen, 1980; Johannessen and Steel,
1992; Braathen et al., 2011; Figure 2). Third, the Moscovian Minkinfjellet Formation is dominated
by limestone and dolomite with minor evaporites (Carronelva and Terrierfjellet members), and
carbonate karst breccias (Fortet Member; McWhae, 1953; Cutbill and Challinor, 1965; Lønøy,
1995; Figure 2). Fourth and fifth, the Wördiekammen and Gipshuken formations mainly consist of
dolomite and limestone interbedded with evaporites and crosscut by dissolution breccias in the
latter (Gee et al., 1952; Cutbill and Challinor, 1965).

By contrast to the pre-rift origin inferred for Mississippian sedimentary units,
Pennsylvanian rocks of the Hultberget, Ebbadalen and Minkinfjellet formations are thought to
represent respectively the early, main and late syn-rift sedimentation episodes (Prosser, 1993) or
the "initiation", "interaction and linkage", and "through-going fault" stages (Gawthorpe and
Leeder, 2000, their fig. 3) in the Billefjorden Trough (Johannessen and Steel, 1992; Braathen et al.,
2011). Pennsylvanian syn-rift sedimentation was accompanied by significant km-scale
downthrowing to the east along the BFZ, and tilting of SW-dipping Carboniferous normal faults
and related fault-propagation folds into a subvertical/east-dipping position in the eastern part of the
Billefjorden Trough (e.g., the Løvehovden fault; Maher and Braathen, 2011; Braathen et al., 2011).
Middle Pennsylvanian–Cisuralian sedimentary strata of the Wördiekammen and Gipshuken
formations are largely accepted as late syn-rift to post-rift sedimentary units (Braathen et al., 2011),
in other words as part of the "through-going fault" stage of Gawthorpe and Leeder (2000). In
Odellfjellet (Austfjorden; Figure 1), newly exposed strata investigated in the present contribution
crop out near cliffs of thick Pennsylvanian sedimentary strata of the Hultberget, Ebbadalen and
Minkinfjellet formations (Johannessen and Steel, 1992; Lamar and Douglaass, 1995).

**Cenozoic fold and thrust belt**

Apart from a few minor tectonic episodes, e.g., in the Permian–Triassic (Worsley and Mørk,
1978; Mørk et al., 1982; Steel and Worsley, 1984; Osmundsen et al., 2014) and potentially in the
Cretaceous (Nemec et al., 1988; Prestholm and Walderhaug, 2000; Onderdonk and Midtkandal,
2010), the Svalbard Archipelago is believed to have remained relatively quiet tectonically from the
Pennsylvanian to the end of the Mesozoic. In mid-Cenozoic times, contractional–transpressional



deformation related to continental break-up and subsequent opening of the Northeast Atlantic
Ocean formed sub-horizontal NW- to NNW-trending folds (Bergh et al., 1997; Bergh and Grogan,
2003), and inverted major normal faults, resulting in the formation of the West Spitsbergen fold-
and-thrust belt (Harland, 1969; Lowell, 1972; Harland et al., 1974; Haremo et al., 1990; Dallmann
et al., 1993; Diβmann and Grewing, 1997). Cenozoic dextral transpression and contraction
reactivated preexisting, margin-parallel N–S-trending Caledonian and margin-oblique NW–SE- to
NNW–SSE-trending Svalbardian (Ellesmerian) folds and thrusts (Bergh et al., 1997; Blinova et
al., 2012, 2013), and inverted Devonian–Carboniferous normal faults such as the BFZ, making
fault offsets difficult to resolve.

## 3. Methods

The present work is a compilation of satellite images from toposvalbard.npolar.no covering
areas in the eastern part of the Billefjorden Trough (Figure 3), and of field structural observations
in Carboniferous sedimentary rocks in Odellfjellet (Figure 1) collected during a field excursion in
summer 2016 (Figure 4). Structural data are plotted in lower-hemisphere, equal-area Schmidt
stereonets as great circles. Satellite images of exposed basement rocks were used to identify brittle
faults in exposed but difficultly accessible Proterozoic basement rocks adjacent to Carboniferous
sedimentary deposits in the Billefjorden Trough.

## 4. Results

**Basement rocks**

East and southeast of the investigated outcrop by a riverbed in Odellfjellet (Figure 1),
Mesoproterozoic to earliest Neoproterozoic basement rocks crop out, and these display a well-
developed N–S-trending gneissic foliation (Harland et al., 1966; Balashov et al., 1993; Witt-
Nilsson et al., 1998; Johansson and Gee, 1999). This prominent ductile fabric is visible on satellite
images where it defines series of clustered, (sub-) parallel, linear to arcuate lineaments following
the topography of ridges exposed within Mittag–Lefflerbreen, e.g., Framstakken (Figure 3a),
Heclastakken (Figure 3b) and Furystakken (Figure 3c), and on mountain flanks, e.g., southernmost
tip of Sederholmfjellet (Figure 3d). In these outcrops, basement rocks are glaciated (Marks and
Wysokinski, 1986) and glacial lineations and features are easily differentiated from basement
ductile fabrics, and correlated with ongoing ice flow (Figure 3; Marks and Wysokinski, 1986).



Discrete, steep, WNW–ESE-trending escarpments occur and trend oblique (sub-orthogonal) to the prominent N–S-trending foliation in Mesoproterozoic to Neoproterozoic basement rocks (Figure 3; Harland et al., 1966; Balashov et al., 1993; Witt-Nilsson et al., 1998; Johansson and Gee, 1999). Further, these escarpments are parallel to steeply dipping strike-slip to

normal brittle faults that crosscut the Atomfjella Antiform in northern Ny-Friesland, e.g., the Mosseldalen fault (Witt-Nilsson et al., 1998). Thus, we interpret the abundant WNW–ESE-trending escarpments in basement rocks in southernmost Sederholmfjellet and in basement ridges in Mittag–Lefflerbreen to represent steep, inherited, Neoproterozoic to early/mid-Paleozoic, WNW–ESE-striking brittle faults. This is supported by outcrop occurrences of similarly striking

basin-oblique brittle faults in Ebbadalen (in Billefjorden; Christophersen, 2015) and Biscahalvøya (in northwestern Spitsbergen; Gee, 1972; Labrousse et al., 2008), which crosscut Mesoproterozoic to earliest Neoproterozoic basement rocks and terminate below unconformably overlying Devonian–Carboniferous sedimentary deposits.

**Sedimentary rocks**

*Dark grey sandstones and coaly shales*

In Odellfjellet (Figure 1 and Figure 4), we evidenced the presence of a several tens of meter thick succession made of meter-thick beds of grey sandstones and dark coaly shales showing a south- to southwest-wards dip (Figure 5a). The lower part of this succession crops out at the river

mouth and is dominated by interbedded, meter-thick beds of coal-bearing shale and grey sandstone (Figure 5a). Coal-bearing shales showed sparse plant fossils, including Stigmaria ficoides (Figure 5b; Playford, 1962; Birkenmayer and Turnau 1962). The upper part of the succession crops out hundreds of meters south- and south-westwards along the riverbed. There, the succession includes in addition beds of grey claystone with iron nodules (Figure 5c) and soil profiles with polygonal

fractures (Figure 5d). One kilometer southwards along the riverbed, the upper part of the succession of grey sandstone–coaly shale crops out again and is interbedded with thin beds of yellow sandstone in the hanging wall of a major fault, the Overgangshytta fault. There, the succession forms a 10–20 meter-wide, E–W- to WNW–ESE-trending, open and upright anticline (Figure 6).

Based on previous description of the Billefjorden Group and Hultberget Formation in

Billefjorden (Cutbill et al., 1976; Gjelberg, 1984), the hereby described grey sandstone and coaly shale sedimentary strata observed at the river mouth and in the hanging wall of the Overgangshytta



fault may either belong to the upper part of the Billefjorden Group or represent the base of the Hultberget Formation. However, iron nodules found in the upper part of the grey sandstone–coaly shale succession have not been described in the lower part of the Hultberget Formation and are

rather typical of the upper part of this Formation (Cutbill et al., 1976). On the contrary, iron nodules are fairly common within the upper part of the Sporehøgda and Birger Johnsonfjellet members of the Mississippian Mumien Formation (Cutbill et al., 1976; Gjelberg, 1984; Figure 2). In addition, the presence of soil profiles (Figure 5d) and Stigmaria ficoides (Figure 5b), a plant fossil abundantly found in the Billefjorden Group (Playford, 1962; Birkenmayer and Turnau 1962;

Gjelberg, 1984), respectively near the top and base of the described outcrops rather suggests that the grey sandstone and coaly shale strata in Odellfjellet are part of the Billefjorden Group.

*Red sandstones and shales*

           At the river mouth, tilted beds of grey sandstones and coal-bearing shales are in angular

unconformity contact with flat-lying beds of red to yellow sandstones partly covered by Quaternary glacial deposits (Figure 7a). Hundreds of meters southwards along the riverbed, grey sandstones and dark coaly shales are interbedded with thin, tens of centimeter-thick beds of yellow sandstone (Figure 7b), which proportion gradually increases southwards. Farther south, coaly shales eventually disappear and are replaced by abundant red sandstone and shale interbedded with

subsidiary grey to yellow sandstone (Figure 7c). Based on the typical red coloration of the dominant sandstone and shale beds and on the presence of thin beds of yellow sandstone and subsidiary grey sandstone (Cutbill et al. 1976; Gjelberg, 1984), we propose that the hereby-described red-bed sedimentary succession is part of the Hultberget Formation (Figure 2).

**Brittle faults**

*Faults within the Billefjorden Group*

           In Odellfjellet (Figure 1 and Figure 4), sedimentary rocks of the Billefjorden Group are crosscut by steep NE–SW- to ENE–WSW-, NW–SE- to WNW–ESE-, and subsidiary NNE–SSW- to N–S-striking faults (Figure 4). Brittle faults display abundant, centimeter- to decimeter-thick

lenses of light-colored, non-cohesive fault-rock (Figure 8a). Slickensides (grooves) along these faults indicate dominant normal dip-slip and subordinate normal oblique-slip movements (Figure 4). WNW–ESE- to NW–SE-striking faults generally die out within grey sandstones and coaly



shales of the Billefjorden Group and often display thickened sandstone beds in the hanging wall, which do not appear to continue into the faults footwall (Figure 8b–c). Based on the dominant

normal sense of shear of these fault, we argue that thickened sedimentary strata in the hanging wall represent potential growth strata reflecting syn-tectonic sedimentation. Notably, Figure 8c shows that, in places, interpreted syn-tectonic growth strata along NNE-dipping faults are composed of two discrete sedimentary units, including proximal sandy wedges and distal prograding to sheet-like sand bodies eroded upwards, which are separated from each other by an angular unconformity.

In places, high-angle brittle faults appear to flatten and sole into shale-dominated beds of the Billefjorden Group, forming duplex-like geometries that incorporate lenses of squeezed shale and cataclasite with clasts of partially preserved coaly shale, as well as possible shallow-dipping bedding-parallel decollements (Figure 8d–e). In cross-section, these flattening brittle faults display normal sense of shear (red line in Figure 8d–e), while smaller faults within duplex-like structures

show minor reverse offsets of host-rock clasts (dashed red lines in Figure 8d–e). We tentatively interpret these as Carboniferous normal faults and duplexes soling downwards into shale-dominated decollements, which were subsequently partly reactivated as reverse faults, possibly during Cenozoic transpression.

*Faults within the Hultberget Formation*

Sedimentary rocks of the Hultberget Formation are crosscut by steep NNE–SSW- to N–S-, NE–SW- to ENE–WSW-, and subsidiary low-angle WNW–ESE-striking faults (Figure 4). Fault-cores include centimeter- to decimeter-thick lenses of non-cohesive light-colored fault-rock (Figure 9a). Displacement along these faults is in the order of a few decimeters to 1–2 meters, as

shown by normal offsets of red and grey sedimentary beds (Figure 9b–d). A major difference between faults crosscutting the Billefjorden Group and those truncating red and grey strata of the Hultberget Formation is that we did not identify any growth strata in the latter, therefore suggesting that movement along brittle faults crosscutting the Hultberget Formation occurred after sediment deposition.


*The Overgangshytta fault*

The southernmost outcrops along the riverbed are crosscut by a major NNE-dipping fault that we name the Overgangshytta fault (Figure 4 and Figure 10a). In the hanging wall, this fault is



characterized by a decametric/mesoscale anticline incorporating beds of grey sandstones and coaly
shales of the Billefjorden Group interbedded with thin beds of yellow sandstone more typical of
the Hultberget Formation (Figure 6). The footwall of the fault is dominated by red sandstones and
shales interbedded with grey to yellow sandstones (Figure 10a). These rocks are similar to those of
the Hultberget Formation farther north along the riverbed (Figure 7c) and to red Devonian
sandstones also observed in the area, west of the BFZ (McCann and Dallmann, 1996). The 2–3
meter-thick fault-core is made of steeply SSW-tilted strata (Figure 10a) crosscut by abundant
fractures comprising centimeter- to decimeter-scale lenses of yellow (Figure 10b) and light-colored
non-cohesive fault-rocks (Figure 10c). The fault shows slickenside lineations indicating dip-slip
normal movements (Figure 10d). The Overgangshytta fault was not observed in adjacent cliffs to
the WNW, where sedimentary strata of the Hultberget, Ebbadalen and Minkinfjellet formations
crop out, possibly suggesting that the fault dies out laterally and/or vertically (Figure 10e and
supplements).

## 5. Discussion

### 5.1. Origin of the Overgangshytta fault

The Overgangshytta fault was not observed in adjacent cliff-outcrops made of sedimentary
strata of the Hultberget, Ebbadalen and Minkinfjellet formations (Figure 4, Figure 10e, and
supplements), thus suggesting that the fault dies out laterally ca. 300 meters to the west-northwest
and/or upwards within the Hultberget Formation. The width of the fault-core (2–3 meters) and the
intensity of deformation in the hanging wall of the fault along the riverbed (Figure 6 and Figure
10a–c) do not support a nearby lateral termination of the fault. However, northwards, along the
riverbed, NNE-dipping faults striking parallel to the Overgangshytta fault die out upwards in coal-
bearing sedimentary rocks of the Billefjorden Group (Figure 8b–c). We therefore propose that the
Overgangshytta fault also dies out upwards within uppermost Mississippian–Lower Pennsylvanian
strata of the Hultberget or Ebbadalen Formation. Such upwards dying-out geometry was also
observed for similarly striking, steep, SW- to SSW-dipping faults in Billefjorden, the
Kampesteindalen fault and Ebbabreen faults. The former dies out within the Ebbadalen Formation
and juxtaposes sedimentary strata of the Hultberget Formation in the footwall with rocks of the
Ebbadalen Formation in the hanging wall (Braathen et al., 2011; Smyrak-Sikora pers. comm.,
2016), whereas the latter downthrow thickened Mississippian rocks of the Billefjorden Group to





the southwest and die out upwards within the Hultberget Formation (McCann and Dallmann, 1996). Thus, the steep and upwards dying-out geometry of the Overgangshytta fault (Figure 10e) together with slickengrooves indicating normal dip-slip movement (Figure 10d) suggest that this fault formed as an extensional normal fault in the Mississippian to earliest Pennsylvanian.

       The red sandstones and shales interbedded with grey to yellow sandstones in the footwall
of the Overgangshytta fault (Figure 10a, d and e) are similar to km-thick Devonian sedimentary deposits observed west of the BFZ in adjacent onshore areas in André Land (Manby and Lyberis, 1992), and their presence in the footwall of the Overgangshytta fault may indicate hundreds of meter- to km-scale, down-NNE, normal displacement along this fault. However, such Devonian deposits have never been observed east of the BFZ and are believed to have been eroded or never
deposited. Thus, sedimentary strata in the footwall of the Overgangshytta fault (Figure 10a and e) are more likely to represent uppermost Mississippian–lowermost Pennsylvanian strata of the Hultberget Formation, analog to those observed in the hanging wall of the fault (Figure 7a–b). Isopach maps from Cutbill et al. (1976) suggest that the Hultberget Formation is no thicker 80 m in Odellfjellet, and, therefore, the presence of sedimentary strata of the Hultberget Formation on
both sides of the Overgangshytta fault may indicate overall vertical displacement < 80 m along the fault.

       East and southeast of the studied outcrops in Odellfjellet (Figure 1), satellite images show numerous WNW–ESE-trending escarpments in Paleoproterozoic to earliest Neoproterozoic basement rocks in Sederholmfjellet and Mittag–Lefflerbreen (Figure 3), which we interpreted as
steep brittle faults. Similar, steep and abundant, WNW–ESE-striking, margin-oblique brittle faults were mapped on the Varanger Peninsula (Siedlecka and Siedlecki, 1967; Siedlecki, 1975, 1980) and Magerøya (Koehl et al., submitted) in northern Norway, and represent fault segments of a major, inherited, Neoproterozoic subvertical fault, the Trollfjorden–Komagelva Fault Zone, which formed during the Timanian Orogeny and is thought to have accommodated hundreds of kilometers
of lateral displacement (Rice, 2013). This fault experienced multiple episodes of reactivation and was last reactivated under transtension in Mississippian times (Visean; Lippard and Prestvik, 1997), shortly before it was intruded by dolerite dykes that seal the fault (Roberts et al., 1991; Nasuti et al., 2015). Hence, we propose that the WNW–ESE-trending fault-related escarpments observed in Paleoproterozoic to earliest Neoproterozoic basement rocks in Sederholmfjellet and
Mittag–Lefflerbreen (Figure 3) correspond to inherited Neoproterozoic (Timanian?) strike-slip



faults. Possible inherited Timanian fabrics also exist in southern Spitsbergen and include steep WNW–ESE- to NW–SE-striking Neoproterozoic faults and shear zones that show affinities with the Timanides of northern Norway (Mazur et al., 2009; Majka et al., 2010), thus supporting our interpretation. Moreover, a recent seismic study suggests a Timanian origin for the WNW–ESE-

trending Olga Basin in the northern Barents Sea (Klitzke et al., submitted). We propose that steep basement-seated margin-oblique faults in central Spitsbergen were partly reactivated as normal faults during post-Caledonian extension and may have localized the formation of Mississippian– earliest Pennsylvanian basin-oblique WNW–ESE-striking normal faults like the Overgangshytta fault in Odellfjellet. Such interpretation accounts both for the strike-slip (inherited?) and normal

(post-Caledonian reactivation?) shear senses inferred for WNW–ESE-striking faults in northern Ny-Friesland (Witt-Nilsson et al., 1998).

In the hanging wall of the Overgangshytta fault, the anticline involving sedimentary rocks of the Hultberget Formation and Billefjorden Group (Figure 6) may represent a normal fault-related fold (Schlische, 1995), e.g. a rollover anticline formed as a response to large extensional

displacement along a listric fault, or a growth anticline formed during the propagation of the fault into overlying sedimentary rocks of the Billefjorden Group (?) and Hultberget Formation. An origin as a rollover anticline is incompatible with the inferred geometry of the Overgangshytta fault at depth, as this fault may have formed along (a) preexisting steep–subvertical inherited Neoproterozoic fault(s) and is unlikely to be listric. This is supported by satellite images showing

numerous steep WNW–ESE-trending fault-related escarpments in exposed Paleoproterozoic to earliest Neoproterozoic basement rocks southeast (Mittag–Lefflerbreen; Figure 3a–c) and east of Odellfjellet (Sederholmfjellet; Figure 3d), which most likely continue below the studied outcrops of Carboniferous sedimentary rocks, and by field mapping of abundant steep WNW–ESE-striking faults in northern Ny-Friesland (Witt-Nilsson et al., 1998). Conversely, a formation as a potential

growth anticline is compatible with the inferred steep geometry of the Overgangshytta fault at depth. The Overgangshytta fault may have propagated upwards from an existing, steep, inherited, Neoproterozoic, basement-seated fault during post-Caledonian Mississippian to earliest Pennsylvanian extension. Such mechanism was recently proposed to explain the geometry of the N–S-striking Løvehovden fault in Billefjorden (Maher and Braathen, 2011). Another possibility is

that the Overgangshytta anticline formed as a fault-bend anticline (Rotevatn and Jackson, 2014,




their fig. 4b) during downward linkage of the Overgangshytta fault with a preexisting basement-seated WNW–ESE-striking fault during (Late/latest?) Mississippian–Pennsylvanian extension.

Alternatively, the observed anticline (Figure 6) formed much later, during Cenozoic contraction–dextral transpression associated with the formation of the West Spitsbergen fold-and-thrust belt (Harland, 1969; Lowell, 1972; Bergh et al., 1997; Leever et al., 2011), thus potentially reflecting top-SSW thrusting. The Overgangshytta fault actually strikes subparallel to most NW–SE-striking Cenozoic thrust faults mapped onshore western Spitsbergen (Braathen and Bergh, 1995; Bergh et al., 1997, 2000) and in nearshore fjords in central Spitsbergen (Bergh et al., 1997; Blinova et al., 2012, 2013). Considering its obliquity with the main N–S- to NNW–SSE-trending axis of the West Spitsbergen fold-and-thrust belt, the Overgangshytta fault might have reactivated as a minor oblique thrust fault during a stage of dextral transpression. This is consistent with minor reverse offsets in small-scale duplexes localized within bedding-parallel decollement levels in shale-dominated beds of the Billefjorden Group in Odellfjellet, which might represent minor inversion of Carboniferous normal faults during Cenozoic transpression (Figure 8d–e). Moreover, analog field studies along fault segments of the San Andreas fault in Indio Hills (Koehl et al., 2017, unpublished) and Mecca Hills in California (Bergh et al., submitted) show that minor thrust faults developed oblique to major strike-slip faults during dextral transpression, and the relative orientation of these oblique thrusts compared to the San Andreas fault matches that of the Overgangshytta fault compared to the BFZ in Svalbard (Figure 1 and Figure 4).

Despite having potentially reactivated as a minor oblique thrust during Cenozoic dextral transpression, the Overgangshytta fault did not propagate into adjacent cliff-outcrops made of Pennsylvanian deposits (Figure 10e, and supplements). We argue that this may be ascribed to the observed steep and inferred subvertical geometries of the Overgangshytta fault at surface and at depth respectively, which were most likely not suitable to accommodate significant reverse displacement (as observed for small-scale duplexes; Figure 8d–e). As a result, the fault was only mildly reactivated with little or no upwards propagation, and adjacent sedimentary rocks of the Hultberget Formation and Billefjorden Group were gently folded (Figure 6). Alternatively or in addition, low-angle bedding-parallel decollements in shaly beds of the Billefjorden Group might have inhibited Cenozoic deformation, thus explaining the lack of inversion structures in the studied outcrops, and resulting in mild inversion of the Overgangshytta fault (Figure 10a) and duplex-like geometries and minor reverse faulting in Mississippian shales (Figure 8d–e). Noteworthy, the



Overgangshytta anticline might as well be the result of combined Carboniferous normal fault-related folding and Cenozoic inversion.

### 5.2. Mississippian extension

*Mississippian growth strata along basin-oblique faults*

Evidence in favor of Mississippian syn-sedimentary extensional brittle faulting include fault slickenside lineations yielding dominant normal dip-slip and subsidiary normal oblique-slip sense of shear (Figure 4 and Figure 10d), and thickened sedimentary beds interpreted as fault-growth strata in the hanging wall of NNE-dipping brittle faults crosscutting coal-bearing sedimentary rocks of the Billefjorden Group (Figure 8b and c). Although it was not possible to measure the strike of the faults showing Mississippian growth strata in the hanging wall, they obviously trend sub-parallel to the NNE-dipping Overgangshytta fault (Figure 4, and Figure 8b–c). Importantly, in Figure 8c, the interpreted syn-tectonic unit in the hanging wall of the NNE-dipping fault displays a proximal sandy wedge and an onlapping, distal, prograding to sheet-like sand body. On the one hand, based on the thickening of the wedge towards the fault and on intra-bedding surfaces (dotted yellow lines in Figure 8c), the proximal sand-rich wedge is believe to reflect a period of normal faulting with rapid accommodation creation (Osmundsen et al., 2014, their fig. 12a). Mississippian normal faulting in Austfjorden is also supported by dominant WNW–ESE- to NW–SE-trending paleocurrent data from the Sporehøgda Member in Lemstrømfjellet (Figure 1), on the eastern shore of Austfjorden (Gjelberg, 1981; his fig. 4.5), suggesting that sedimentary strata of the Sporehøgda Member, both in Odellfjellet and Lemstrømfjellet, might have deposited along active WNW–ESE-striking faults.

On the other hand, the geometry of the distal prograding to sheet-like sand body in Figure 8c suggests a period of slow accommodation creation (Osmundsen et al., 2014, their fig. 12c and d), potentially reflecting upward propagation of the fault as a blind fault, as shown in Gawthorpe et al. (1997, their fig. 3a) and as inferred for the Løvehovden fault farther south, in Billefjorden (Maher and Braathen, 2011), and, thus, indicating decreasing fault activity along WNW–ESE-striking faults during the deposition (of the upper part?) of the Sporehøgda Member (Mumien Formation, Billefjorden Group) in Odellfjellet. Unlike the Overgangshytta fault, minor WNW–ESE-striking faults displaying growth strata in cross-section do not extend upwards into red beds of the Hultberget Formation (Figure 8b–c). This suggests that extensional faulting along WNW–



ESE-striking faults ceased prior to the late Serpukhovian (latest Mississippian), which is consistent with the tectono-sedimentary interpretation of intra-growth-strata packages along these faults that indicate decreasing extension (Figure 8c). However, this does not necessarily imply that regional extension ended in the Mississippian, and shallow-dipping bedding-parallel duplex-shaped decollements in (coaly) shale-dominated beds of the Billefjorden Group in Odellfjellet may have partly decoupled extensional deformation, potentially preventing further (Pennsylvanian) movements along margin-oblique WNW–ESE-striking faults (Figure 8c–e).

Nevertheless, the minimum (Late/latest?) Mississippian age of WNW–ESE-striking faults in Odellfjellet is consistent with Mississippian (Visean) $^{40}$Ar–$^{39}$Ar ages obtained on dolerite dykes intruded during extension/transtension and sealing segments of the Trollfjorden–Komagelva Fault Zone in northern Norway (Roberts et al., 1991; Lippard and Prestvik, 1997). It is also consistent with Late Devonian–Mississippian K–Ar ages obtained for fault gouge in northern Norway (Davids et al., 2013; Torgersen et al., 2014; Koehl et al., 2018b) and northeast Greenland (Rotevatn et al., 2018). This also possibly suggests that the Overgangshytta fault initially died out within Mississippian strata of the Billefjorden Group and, later on, propagated into overlying sedimentary deposits of the Hultberget Formation, potentially during a mild episode of inversion of the fault during Cenozoic contraction–transpression. As proposed for the Overgangshytta fault, it is probable that most WNW–ESE-striking normal faults described in the present study formed along reactivated basement-seated Neoproterozoic fabrics (Figure 3).

By contrast, although showing meter-scale normal offsets and slickenside lineations indicating normal sense of shear (Figure 4 and Figure 9), N–S- and NE–SW-striking faults observed in Mississippian–lowermost Pennsylvanian strata of the Billefjorden Group and Hultberget Formation along the riverbed in Odellfjellet (Figure 8a and Figure 9b–d) did not display evidence of growth strata. Hence, the timing of formation of these faults remains uncertain. Nevertheless, knowing that the study area (Odellfjellet; Figure 1 and Figure 4) and, conceivably, most areas in central Spitsbergen were subjected to tectonic extension in the (Late/latest?) Mississippian (Figure 8b–c and Figure 10d), we propose that N–S- and NE–SW-striking faults (at least some of them) formed and acted simultaneously with WNW–ESE-striking faults during Mississippian extension, the only difference being that faults of the former two trends (N–S- and NE–SW-) experienced further normal movement, possibly during (Early–Middle?) Pennsylvanian



extension (Braathen et al., 2011), thus crosscutting rocks of the Hultberget Formation (Figure 8a and Figure 9b–d).

*Tilting of Mississippian strata of the Billefjorden Group*

In the north, sedimentary strata of the Billefjorden Group appear tilted and dip gently to the southwest, forming an angular unconformity with overlying flat-lying red-beds of the Hultberget Formation (Figure 7a). In the south, grey sandstones and coal-bearing sedimentary rocks of the
500 Billefjorden Group are interbedded with and gradually replaced by conformably overlying clastic redbeds of the Hultberget Formation (Figure 6 and Figure 7c). We argue that the observed angular unconformity in the north represents the distal portion of an uplifted, partly exposed rotated fault-block, and that conformably overlying beds of the Billefjorden Group and Hultberget Formation farther south correspond to proximal, hanging wall, syn-tectonic sedimentary strata deposited in a
505 constantly or repeatedly flooded portion of an active fault-block (Figure 11). Consequently, the southwestward tilting of Mississippian sedimentary strata may reflect (Late/latest?) Mississippian extensional faulting along (a) NNE- to NE-dipping brittle fault(s), possibly the Overgangshytta fault and/or (a) similarly trending and dipping fault(s), e.g., Figure 8b and c, thus supporting that extension initiated prior to the deposition of red-colored sedimentary strata of the Hultberget
Formation. This interpretation is supported by similar observations in western Spitsbergen, where Mississippian coal-bearing sedimentary strata were proposed to have deposited in the hanging wall of an active SSW-dipping normal fault located in Kongsfjorden, forming a WNW–ESE-trending Mississippian basin, the Brøggerhalvøya trough (Bergh et al., 2000). The absence of Mississippian sedimentary strata northeast of Brøggerhalvøya was ascribed to uplift and erosion of the footwall
of the fault in Kongsfjorden, and the fining upwards pattern recorded in the strata suggested to represent a break in normal faulting activity near the end of the Mississippian (Fairchild, 1982).

Furthermore, in the Barents Sea, a major Late Mississippian (Serpukhovian) unconformity was described onshore Bjørnøya (Worsley et al., 2001) and on the Finnmark Platform (Bugge et al., 1995; Koehl et al., 2018a). This unconformity was correlated to a major eustatic sea-level fall
at ca. 330 Ma (Saunders and Ramsbottom, 1986; Haq and Schutter, 2008). This short-lived eustatic sea-level fall was followed by eustatic sea-level rise at ca. 325 Ma (late Serpukhovian; Saunders and Ramsbottom, 1986; Haq and Schutter, 2008) coinciding with the deposition of the Hultberget Formation (Cutbill and Challinor, 1965). In Odellfjellet, the local absence of Late Mississippian



unconformity indicates that parts of central Spitsbergen remained flooded through the
Serpukhovian, and these flooded areas appear to be located in the hanging wall of NNE-dipping
faults (e.g., the Overgangshytta fault) that accommodated normal displacement in the (Late/latest?)
Mississippian (Figure 11). Thus, it is possible that areas where beds of the Hultberget Formation
conformably overlie Mississippian strata of the Billefjorden Group, like in Billefjorden (central
Spitsbergen; Cutbill et al., 1976) and Ditlovtoppen (eastern Spitsbergen; Scheibner et al., 2015),
represent proximal portions of hanging walls (i.e., located near the fault) that were down-faulted
during active normal faulting in the (Late/latest?) Mississippian.

Alternatively, tilting of Mississippian strata in Odellfjellet might originate from Late
Devonian–Mississippian (Ellesmerian) and/or Cenozoic transpression. However, Late Devonian–
Mississippian transpression does not reconcile the interbedded character of the Hultberget
Formation and Billefjorden Group, which conformably overlie one another in the south (Figure 6,
and Figure 7b–c), and Cenozoic transpression would have resulted in the folding of the
unconformity between the Billefjorden Group and Hultberget Formation in the north. Another
explanation might be along-strike variation in displacement magnitude along the BFZ during the
deposition of sedimentary strata of the Billefjorden Group, resulting in so-called "transverse folds"
(Schlische, 1995). However, on the Finnmark Platform in the SW Barents Sea, Mississippian strata
appear tilted along brittle normal faults and are partially eroded in distal portions of hanging walls
(e.g., Koehl et al., 2018a, their fig. 6a). Thus, we favor an interpretation related to down-NNE
normal faulting for the observed southwestwards tilting of Mississippian sedimentary strata in
Odellfjelllet (Figure 11).

*Switch from widespread to localized extension*

Our observations in Odellfjellet show that basin-oblique, WNW–ESE- to NW–SE-striking
normal faults were active in (until?) the (Late/latest?) Mississippian (Figure 8b–c). Similarly, in
Birger Johnsonfjellet (central Spitsbergen), N–S-striking faults showing growth strata with syn-
depositional tilting die out upwards within Mississippian deposits of the Billefjorden Group
(McCann and Dallmann, 1996), thus suggesting that at least some N–S-striking faults were active
during Mississippian extension. Thus, we propose that central Spitsbergen was subjected to
widespread Mississippian extension distributed along numerous faults of varied trends, including
margin-oblique WNW–ESE- to NW–SE- (Figure 8b–c and Figure 10a) and margin-parallel N–S-



striking faults (McCann and Dallmann, 1996), and, conceivably, NE–SW-striking faults, thus possibly representing the rift "initiation" phase as detailed in Gawthorpe and Leeder (2000).

Margin-oblique faults systematically die out upwards within Mississippian (to lowermost Pennsylvanian) strata in Odellfjellet (Figure 8b–c), Billefjorden (e.g., Ebbabreen and Kampesteindalen faults; McCann and Dallmann, 1996; Braathen et al., 2011; Smyrak-Sikora pers.
comm., 2016) and Bjørnøya (e.g., Russleva fault; Braathen et al., 1999; Koehl, in prep.). In addition, inherited margin-oblique faults in northern Norway were dated to have been last active in the Mississippian (e.g., Trollfjorden–Komagelva Fault Zone; Lippard and Prestvik, 1997). By contrast, only a few margin-parallel (N–S-striking) faults die out within Mississippian sedimentary deposits in central Spitsbergen (McCann and Dallmann, 1996), while most of these (e.g., BFZ;
Harland et al., 1974; Braathen et al., 2011) and NE–SW-striking faults (Figure 8a and Figure 9b–d) cut through Pennsylvanian sedimentary rocks, suggesting that they remained active through the Early–Middle Pennsylvanian. We therefore propose that central Spitsbergen was subjected to an episode of continuous (Late/latest?) Mississippian–Middle Pennsylvanian extension during which normal displacement progressively localized along fewer fault trends (N–S and NE–SW;
"interaction and linkage" phase of Gawthorpe and Leeder, 2000) , possibly using shallow-dipping bedding-parallel decollements in (coaly) shale-dominated beds of the Billefjorden Group (Figure 8d–e) to decouple margin-oblique WNW–ESE-striking faults. Eventually, extension localized along a few major faults, such as the BFZ ("through-going fault" phase of Gawthorpe and Leeder, 2000), before ultimately ceasing in the Middle–Late Pennsylvanian.

This is similar to what was observed in the southwesternmost Nordkapp basin and on the Finnmark Platform in the SW Barents Sea (Koehl et al., 2018a), where thickened Mississippian sedimentary deposits and adjacent and/or underlying basement rocks are crosscut and offset by numerous normal faults showing mostly minor offsets, whereas thickened wedges of syn-tectonic Pennsylvanian deposits are observed exclusively in the hanging wall of a few major normal faults
displaying hundred meter- to kilometer-scale offsets (e.g., the Langfjorden–Vargsundet fault; Koehl et al., 2018a). Similarly, a switch from widespread extension with multiple active faults accommodating small amount of normal displacement (with slow slip rates) during a phase of rift "initiation", to extension localized along a few major fault surfaces (with high slip rates) during "interaction and linkage" to "through-going fault" phases was also suggested for Jurassic rifting in



the North Sea, where the high-slip rate Gullfaks–Visund fault (Cowie et al., 2005) may represent a younger offshore analog to the BFZ.

        Furthermore, in the NW Barents Sea, a recent seismic study shows thick packages of high-amplitude, south- to southwest-dipping reflections within the Capria Ridge, on the northern flank of the Sørkapp depression (Anell et al., 2016, their figure 3a). These are similar to thick seismic

packages in the SW Barents Sea (Koehl et al., 2018a) and North Sea (Phillips et al., 2016; Fazlikhani et al., 2017) potentially representing inverted Caledonian shear zones. In the NW Barents Sea, these thick packages of high-amplitude reflections are disrupted by (sub-) parallel (i.e., E–W- to NW–SE-striking), margin-oblique, high-angle brittle normal faults, displaying thick wedges of potential Devonian (?) to Mississippian sedimentary rocks in the hanging wall. These

E–W- to NW–SE- striking normal faults mostly die out near the base of a thin overlying layer of (uppermost?) Pennsylvanian sedimentary deposits showing relatively constant thickness (Anell et al., 2016). Hence, extensive normal faulting and thickened sedimentary wedges (growth strata?) along deep, margin-oblique, E–W- to NW–SE-striking faults in the NW Barents Sea, suggest extensive (collapse-related?) extension in Devonian (?) – Mississippian times and decreasing

extension in the Pennsylvanian, which is consistent with field observations in Odellfjellet (Figure 8b–c). Decreasing extension in the Pennsylvanian is also supported by field observations in central Spitsbergen, suggesting that transgression–regression cycles in Pennsylvanian–Cisuralian deposits were mostly controlled by eustatic sea-level changes and only moderately by active faulting along margin-parallel faults like the BFZ (Samuelsberg and Pickard, 1999).

A WNW–ESE to NW–SE direction was proposed for late Paleozoic extension along the Lofoten–Vesterålen and SW Barents Sea margins in northern Norway (Bergh et al., 2007; Hansen et al., 2012; Indrevær et al., 2013). We therefore believe that Spitsbergen was subjected to a similarly oriented stress field rather than the ENE–WSW extension direction proposed by McCann and Dallmann (1996). We argue that WNW–ESE- to NW–SE-directed late Paleozoic extension in

central Spitsbergen may explain the observed upwards dying-out geometry of unsuitably oriented, inherited, basin-oblique, WNW–ESE- to NW–SE-striking faults, while N–S- and NE–SW-striking faults accommodated further (Early–Middle) Pennsylvanian extensional faulting.

        A major difference between margin-oblique faults in Odellfjellet (central Spitsbergen) with their counter parts in northern Norway is that the latter accommodated dominantly lateral post-

Caledonian (transfer) movement, e.g., the Trollfjorden–Komagelva Fault Zone (Koehl et al.,




submitted), whereas the former accommodated dominantly normal dip-slip to oblique-slip motions (Figure 4, Figure 8b–c, and Figure 10d). A tentative explanation might be that inherited, Neoproterozoic, WNW–ESE- to NW–SE-striking brittle faults in central Spitsbergen reactivated as transverse faults (Ogata et al., 2014) in or near the crest of transverse folds reflecting differential

displacement along the BFZ (Schlische, 1995), or as accommodation cross faults (Sengör, 1987), as proposed for the WNW–ESE-striking segment of the Troms–Finnmark Fault Complex in the SW Barents Sea (Koehl et al., 2018a). Such interpretations imply that large-scale normal displacement along margin-parallel faults in central Spitsbergen (e.g., the BFZ) initiated in the Mississippian.


## 6. Conclusions

1) Extensional growth strata in the hanging wall of margin-oblique NNE-dipping normal faults, and the change from unconformable to interbedded contact between tilted Mississippian coal-bearing sedimentary rocks of the Billefjorden Group and flat-lying to tilted uppermost

Mississippian–lowermost Pennsylvanian redbeds of the Hultberget Formation towards major margin-oblique faults (e.g., the Overgangshytta fault) suggest that the former represent early syn-rift deposits that were deposited during (Late/latest?) Mississippian extension.

2) WNW–ESE- to NW–SE-striking faults systematically die out upwards within sedimentary strata of the Billefjorden Group and, occasionally, of the Hultberget Formation, thus suggesting

a switch from widespread extension in the Mississippian involving faults of as many as three trends (WNW–ESE, N–S, and possibly NE–SW) during the rift "initiation" phase, to more localized extension in (Early–Middle?) Pennsylvanian times when normal displacement progressively localized along fewer fault trends (N–S and NE–SW) during the "interaction and linkage" phase, and, eventually, along a few major basin-parallel faults(e.g., Billefjorden Fault

Zone) during the "through-going fault" phase, before extension ceased in the Middle–Late Pennsylvanian.

3) In the Carboniferous, central Spitsbergen was probably subjected to WNW–ESE- to NW–SE-directed extension, thus potentially explaining why unsuitably oriented margin-oblique WNW–ESE-striking faults die out within Mississippian–lowermost Pennsylvanian strata of the

Billefjorden Group and Hultberget Formation, while N–S- and NE–SW-striking faults experienced further normal faulting in the Pennsylvanian.



4) The presence of abundant WNW–ESE-striking fault-related lineaments in Proterozoic basement rocks east and southeast of Odellfjellet indicates that the formation of Mississippian basin-oblique WNW–ESE-striking normal faults (e.g., Overgangshytta fault) in the

Billefjorden Trough may have been controlled by existing Neoproterozoic (Timanian?) basement-seated faults, which

5) Basement-seated Neoproterozoic brittle faults possibly reactivated as transverse faults or accommodation cross faults in the crest of transverse folds that reflect differential displacement along the Billefjorden Fault Zone, hence suggesting that normal displacement along major

margin-parallel faults (like the Billefjorden Fault Zone) initiated in the Mississippian.

6) The juxtaposition of rocks of the Billefjorden Group in the hanging wall of the Overgangshytta fault where they form a major anticline with redbeds of the Hultberget Formation in the footwall of the fault possibly indicates that the fault was mildly reactivated as an oblique thrust during Cenozoic transpression–contraction. Alternatively or complementary, kinematic indicators

with normal sense of shear along the fault suggest that the anticline might have initiated as a growth anticline due to upwards propagation of a preexisting basement-seated fault during (Late/latest?) Mississippian to Early–Middle Pennsylvanian extension.

7) Bedding-parallel decollements in gently dipping Mississippian (coaly) shale-dominated beds of the Billefjorden Group potentially decoupled unsuitably oriented margin-oblique WNW–

ESE-striking faults, preventing further (Pennsylvanian) normal movements along these, and, eventually, partially reactivated as duplex-shaped decollements during Cenozoic transpression, largely inhibiting or preventing Cenozoic inversion of steep Mississippian normal faults.

**Author contribution**

JBPK acquired field measurements, wrote most of the text and drafted all the figures. JMMB contributed with broadening the scope of the discussion and parts of the outcrop description, leading to the addition of multiple paragraphs to the manuscript. Contributions are as follows: JBPK (80%) and JMMB (20%).

**Competing interests**

The authors declare that they have no conflicts of interest.



**Acknowledgements**

The present study is part of the ARCEx (Research Centre for Arctic Petroleum
Exploration), which is funded by the Research Council of Norway (grant number 228107) together
with 10 academic and eight industry partners. We would like to thank all the persons from these
institutions that are involved in this project. We also acknowledge the support of the Arctic Field
Grant (grant number 227549) financing helicopter support to and from Odellfjellet. The authors
are also grateful to Prof. Lars Stemmerik (Natural History Museum of Denmark, København), and
Aleksandra Smyrak-Sikora, Prof. Snorre Olaussen and Erik Johannessen (University Centre in
Svalbard) for field collaboration and constructive comments. The authors also thank Luisa
Campiño (Equinor), Prof. Alvar Braathen (University of Oslo) and Winfried Dallmann (UiT The
Arctic University of Norway in Tromsø) for fruitful discussion that contributed to improve the
manuscript.

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




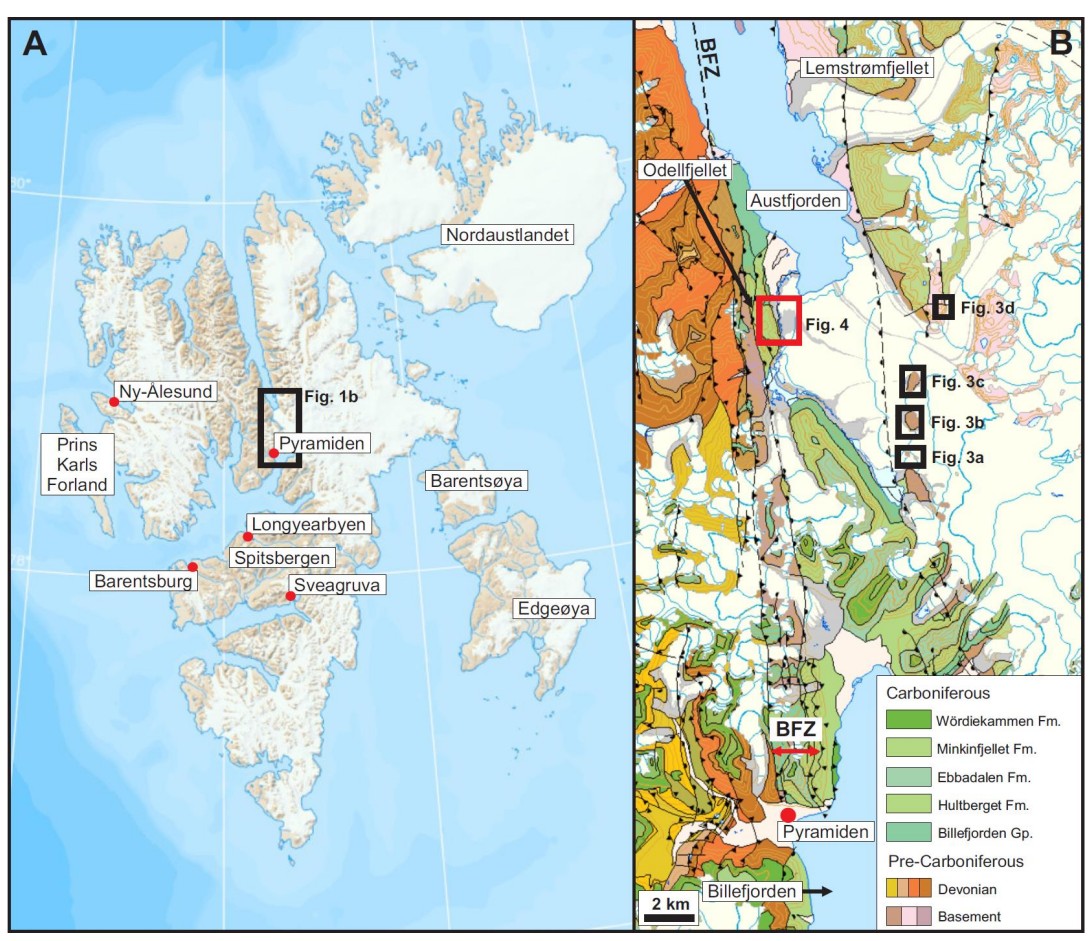

Figure 1: (a) Topography map of Spitsbergen, Svalbard. Modified from toposvalbard.npolar,no; (b) Geological map of the Billefjorden–Austfjorden area, which location is shown in (a). The location of studied outcrops is shown by a red frame. The red arrow shows the width of the Billefjorden Fault Zone (BFZ) at Pyramiden, in Billefjorden. The map is from svalbardkartet.npolar.no.





Figure 2: Lithostratigraphic chart of late Paleozoic sedimentary rocks in central Spitsbergen. The chart is based on descriptions by Gee et al. (1952), McWhae (1953), Playford (1962), Cutbill and Challionor (1965), Holliday and Cutbill (1972), Cutbill et al. (1976), Johannessen (1980), Gjelberg (1981, 1984), Gjelberg and Steel (1981), Johannessen and Steel (1992), Lønøy (1995), Dallmann (1999), Braathen et al. (2011), and Scheibner et al. (2015).



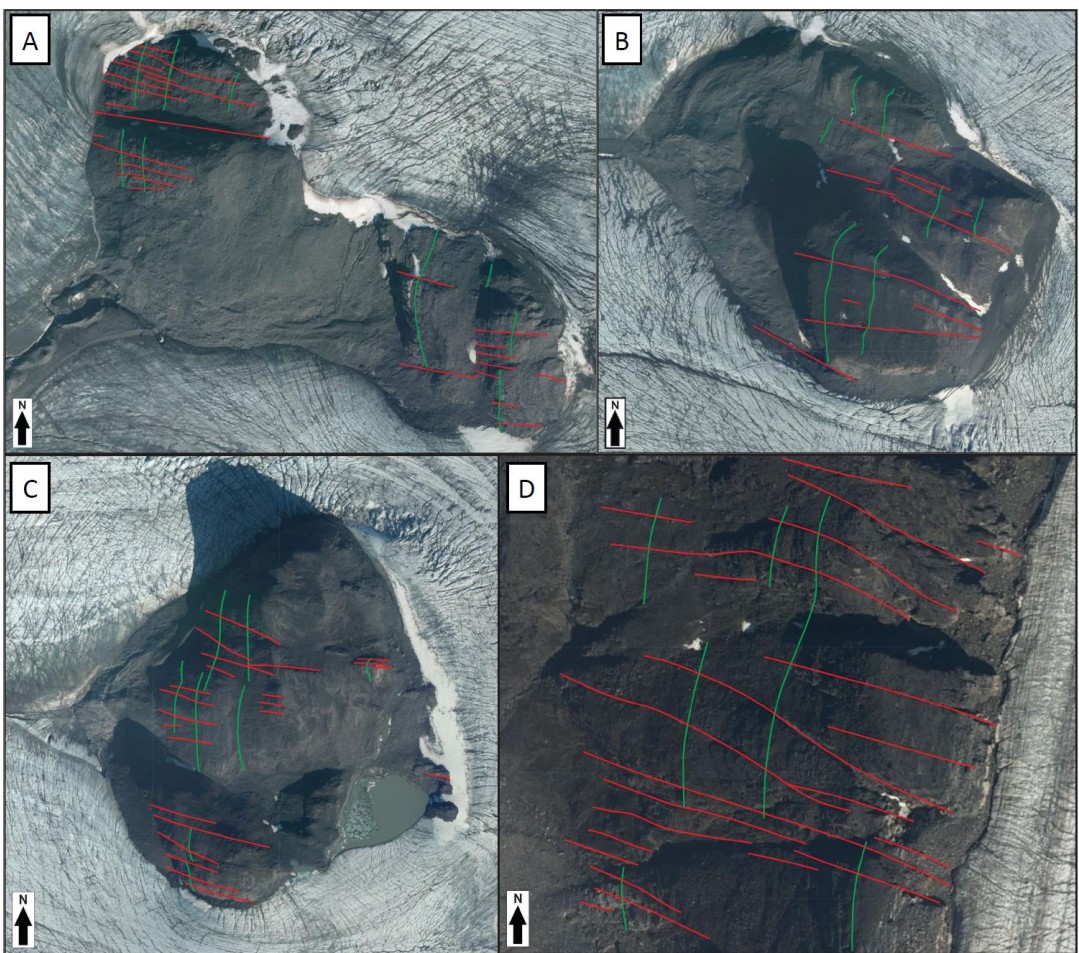

Figure 3: Satellite images from toposvalbard.npolar.no showing arcuate to rectilinear lineaments representing the prominent N–S-trending gneissic foliation of the Atomfjella Antiform and WNW–ESE-trending lineaments interpreted as steep Neoproterozoic brittle faults in basement exposures east and southeast of Odellfjellet, in (a) Framstakken, (b) Heclastakken, (c) Furystakken, and (d) southernmost Sederholmfjellet. Locations are displayed as black frames in Figure 1b. The photographs are approximately one kilometer wide.




Figure 4: Satellite image from toposvalbard.npolar.no showing the study area in Odellfjellet. The studied outcrops are located along a riverbed and consist of sedimentary rocks of the Billefjorden Group (blue double arrows) and Hultberget Formation (pink double arrows) crosscut by brittle faults (e.g., the Overgangshytta fault – Ovf). The riverbed runs sub-parallel to mountain cliff-outcrops made of sedimentary strata of the Hultberget, Ebbadalen and Minkinfjellet formations (Odellfjellet). Dotted orange lines represent stratigraphic boundaries between the Billefjorden Group and Hultberget Formation. Bedding surface measurements as white lines. Stereoplots show (1) great circle fracture surfaces within rocks of the Billefjorden Group and (2) Hultberget Formation, and (3) poles and vectors of slickenside lineations along brittle faults crosscut rocks of the Billefjorden Group and Hultberget Formation. Location is shown by a red frame in Figure 1b.






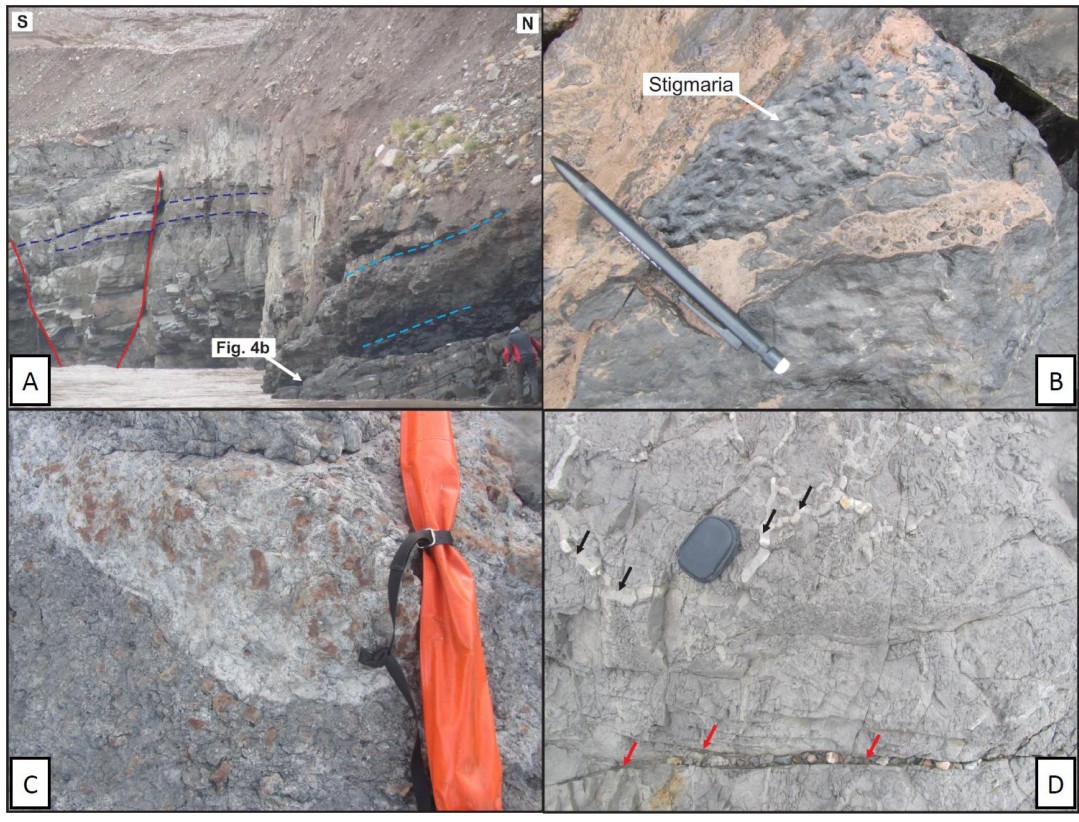

Figure 5: Outcrop photographs showing (a) southwestwardly tilted sedimentary rocks of the Billefjorden Group consisting of
         interbedded coal-bearing shales and grey sandstone in the lower part (light blue), and interbedded coaly shales, grey sandstone, and
         grey claystone with iron nodules in the upper part (dark blue); (b) Stigmaria ficoide in the lower part of the Billefjorden Group
         succession. Location shown in (a); (c) grey claystone with abundant iron nodules in the upper part of the Billefjorden Group
         succession; (d) soil features in grey claystone crosscut by fractures (red arrows) and polygonal fractures (black arrows) in the upper
part of the Billefjorden Group succession.



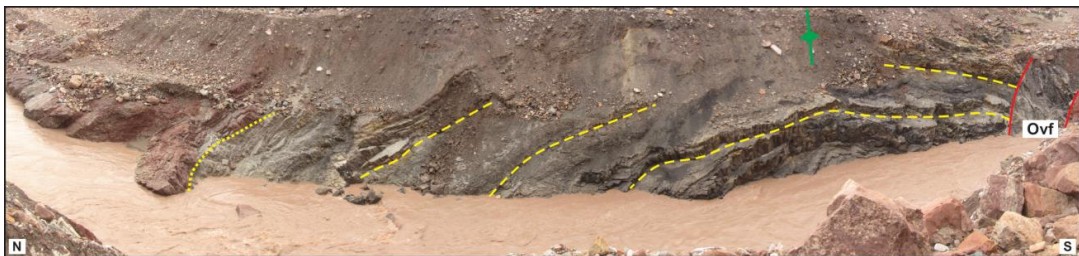

Figure 6: Eastward view of folded sedimentary strata (bedding in dashed yellow) forming an E–W- to WNW–ESE-trending, open, upright anticline (green) in the hanging wall of the Overgangshytta fault (red; Ovf), along the southern portion of the riverbed (Figure 4). Note the boundary (conformity?) between grey sandstones and coaly shales of the Billefjorden Group and red sandstones and shales of the Hultberget Formation in dotted yellow. The outcrop is approximately 10–15 m wide. See green line in Figure 4 for location of the anticline.




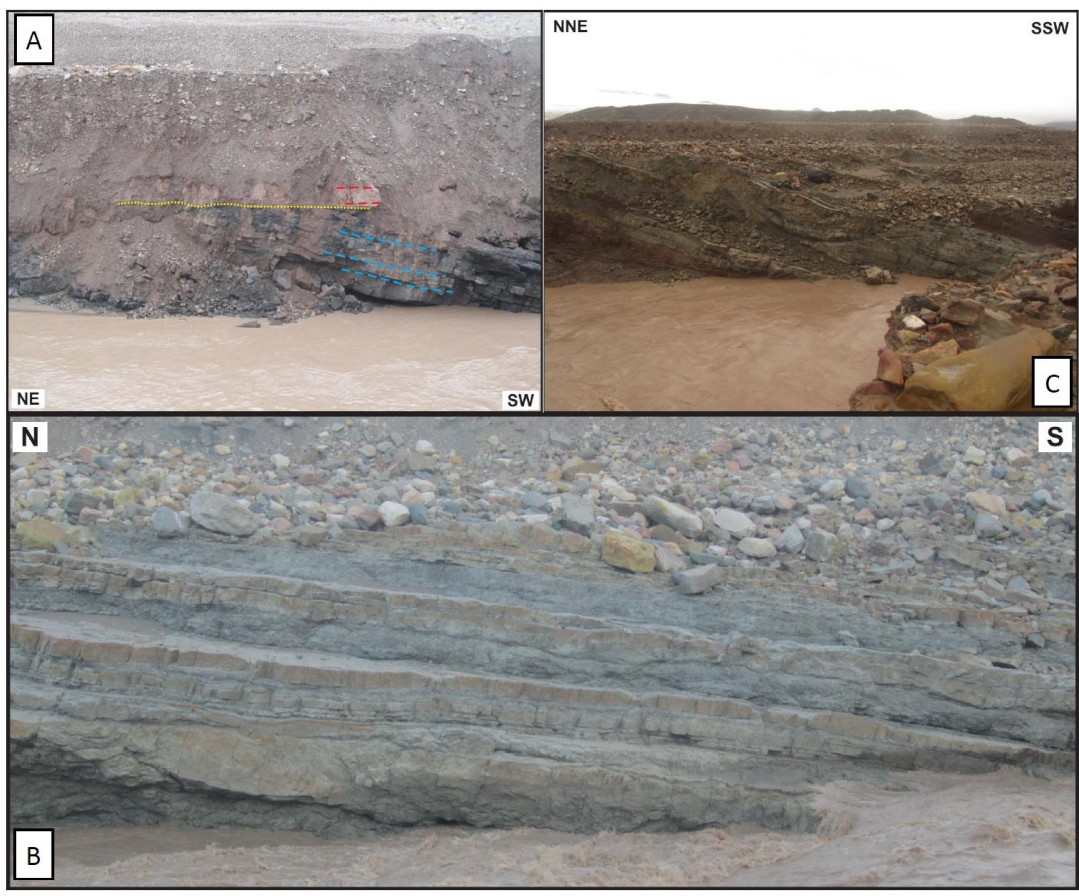

Figure 7: (a) Outcrop photographs showing southwestwardly tilted sandstone- and coaly shale-rich beds of the Billefjorden Group (dashed blue) unconformably (unconformity in dotted yellow) overlain by flat-lying (dashed red) redbeds of the Hultberget Formation. Outcrop located at the northern end of the riverbed in Figure 4; (b) Grey sandstone and shales interbedded with thin beds of yellow sandstones (transitional between Billefjorden Group and Hultberget Formation?); (c) Red shale interbedded with grey and yellow sandstone characteristic of the Hultberget Formation in Odellfjellet, Austfjorden.








Figure 8: Outcrop photographs showing (a) centimeter–decimeter thick lenses of light-colored non-cohesive fault-rock along a brittle fault truncating grey sandstones of the Billefjorden Group; (b) NNE-dipping normal faults showing meter-scale down-NNE movement and potential growth strata (between the green and blue markers). The faults crosscut sandstones and shales of the Billefjorden Group in which they die out upwards. The outcrop width is approximately 5–6 m wide; (c) potential fault-growth strata made of dark sandstone (between the green and orange markers) in the hanging wall of a NNE-dipping brittle fault with decimeter–meter-scale normal displacement. The fault dies out upwards within sedimentary strata of the Billefjorden Group. The interpreted syn-tectonic growth strata is composed of two sedimentary packages, including a proximal sandy wedge thickening towards the fault, and a distal prograding to sheet-like sand rich body onlapping (yellow arrows) the proximal wedge. The two packages are separated by an angular unconformity (dotted white line) and are both eroded upwards (orange dashed line). Yellow dotted lines represent intra-bed surfaces. The outcrop width is approximately 7–8 m wide; (d) Outcrop photograph showing a high-angle brittle normal fault (red line) in grey sandstone of the Billefjorden Group flattening and soling into a gently south-dipping shale-dominated bed (dashed yellow lines) displaying significant thickness variations. The dotted white frame shows the location of (e), and white boxes structural measurements (fault surface in red, and bedding surface in black); (e) Zoomed in photograph showing thickening of the shale-dominated bed (dashed yellow lines) in the footwall of the flattening normal fault (red line), including fine-grained cataclasite and preserved fragments of coaly shale host-rock (black lines) seemingly offset in a reverse top-northwest fashion by small-scale faults that form a duplex-shaped structure (dashed red lines). In the hanging wall, the shale-dominated bed significantly thins and is preserved as a lens of partly squeezed shale (white line) and cataclasite. Location shown by a dotted white frame in (d).

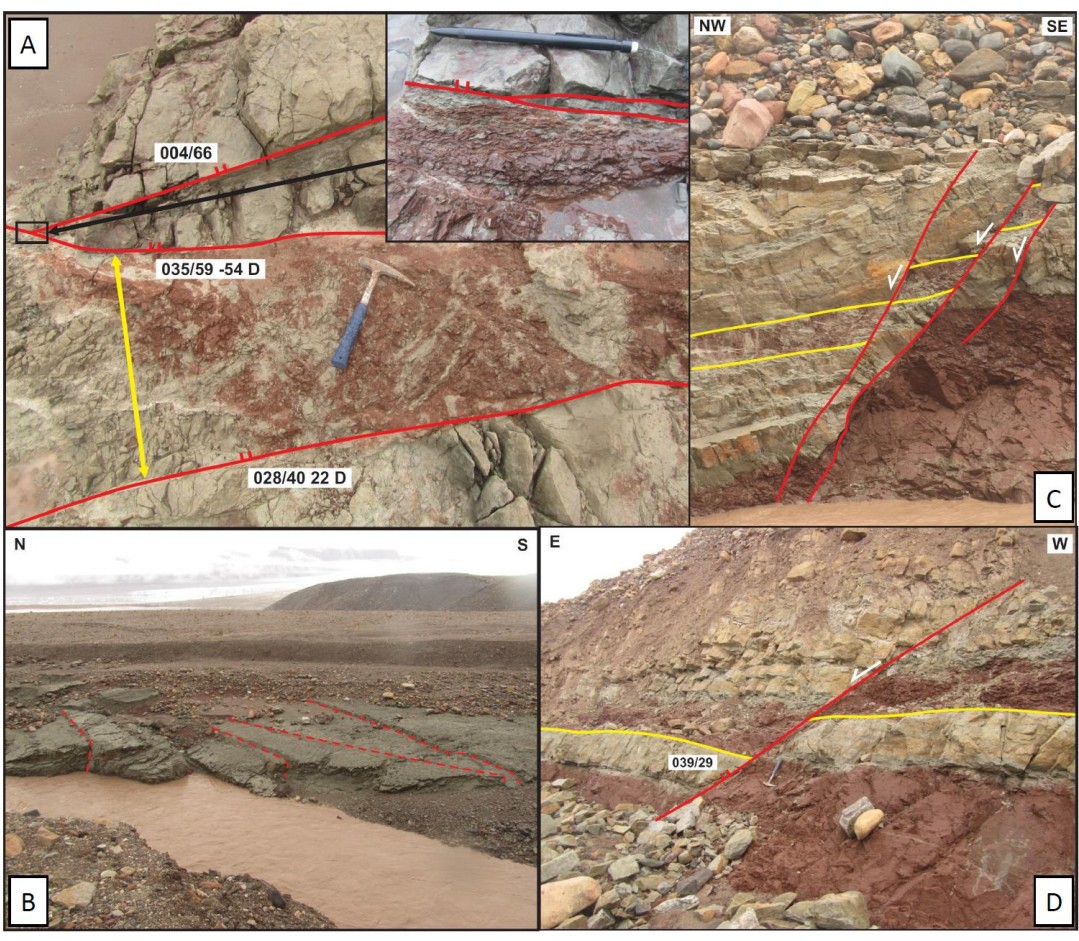

Figure 9: Outcrop photographs along the southern half of the riverbed (Figure 4) showing (a) meter-scale fault-core (yellow arrow) along a SW-dipping fault comprising light-colored and reddish non-cohesive fault-rock derived respectively from adjacent grey sandstone and red shales of the Hultberget Formation. The upper right inset shows shattered sedimentary rocks truncated by numerous sub-parallel brittle shears along the main fault surface; (b) decimeter-scale fault scarps related to decimeter- to meter-scale down-northwest and down-west normal movements along NE–SW- and N–S-striking brittle faults in the Hultberget Formation; (c) decimeter-scale down-NW normal offset (yellow lines) along a NE–SW-striking brittle fault crosscutting red shale and grey sandstone of the Hultberget Formation; (d) meter-scale down-SE offset (yellow lines) along a low angle normal fault truncating the Hultberget Formation.








Figure 10: Outcrop photographs showing the geometry of the Overgangshytta fault in the southern portion of the study area. (a) 2–3 meters wide core (sub-horizontal white arrow, and dashed white lines) of the Overgangshytta fault (red) incorporating meter-size lenses of host-rock (dotted blue). Note the potential meter-scale reverse offset, possibly drag-fold in the hanging wall, and highly tilted (bedding in dashed yellow) character of coaly shales and grey sandstones of the Billefjorden Group across the fault; (b) decimeter-scale light-colored and (c) ca. 10 cm-wide yellowish lenses of non-cohesive fault-rock within the Overgangshytta fault core; (d) slickengrooves and asperities indicating normal dip-slip movement within the Overgangshytta fault core. See Figure 10a






for location; (e) northwestward view of the Overgangshytta fault and adjacent cliff-outcrops made of sedimentary rocks of the Hultberget, Ebbadalen and Minkinfjellet formations suggesting that the fault dies out vertically and/or laterally.



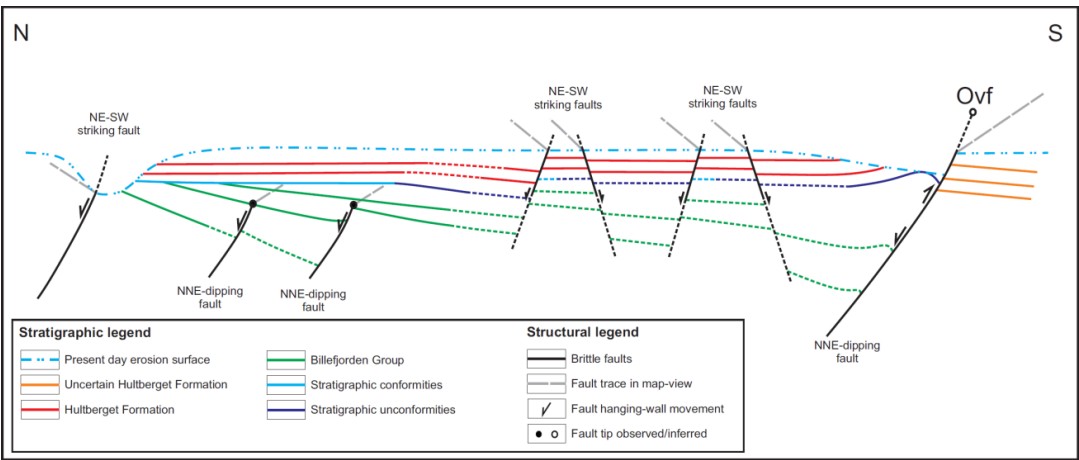

Figure 11: Schematic N–S-oriented cross-section of the studied outcrops in Odellfjellet. The section summarizes observations made along the riverbed and includes upwards dying-out NNE-dipping normal faults with Mississippian growth strata, abundant N–S- and NE–SW-striking normal faults with decimeter–meter-scale offsets, and the Overgangshytta fault (Ovf), a potential Mississippian NNE-dipping normal fault formed along steep, inherited, basement-seated, Neoproterozoic fabrics. The anticline in the hanging wall of the Overgangshytta fault suggests that the fault was inverted as a thrust during Cenozoic transpression. Note

the southward change from unconformity (light blue) to conformity (dark blue) between sedimentary strata of the Billefjorden Group and Hultberget Formation.