# Peer review of "From widespread Mississippian to localized Pennsylvanian extension in central Spitsbergen, Svalbard"

_Solid Earth, 2018_

## Referee Comment (RC1) · A. L. Peace (Referee) · 10 Oct 2018

In their SED paper Koehl and Munoz-Barrera present a study on the structural evolution of central Spitsbergen, Svalbard. In particular, they present new geological field evidence in addition to the analysis of satellite data from the Billefjorden Trough, in Odellfjellet (Austfjorden). The main finding of the work appears to be that the Mississippian sediments of the Billefjorden Group were deposited during active Mississippian extension. Moreover, this deformation is claimed to have been controlled by pre-existing geological structures, a theme addressed in a number of recent SE and SED contributions (Perron et al., 2018; Phillips et al., 2018). SE therefore seems like

an appropriate choice for this work. In addition, I consider this type of field-based study, documenting processes such as reactivation, useful to 'ground truth' and provide analogues for studies interpreting similar processes, which typically use seismic reflection data.

Overall, I think that the work detailed is worthwhile, generally well written and presented. The figures are mostly of good quality and relevant to the manuscript text. My main comments on the manuscript are outlined below, followed by more specific minor suggestions. If the comments outlined can be sufficiently addressed then I would like to recommend publication in SE.

1) Wider implications of the study and comparison to other regions

The authors present an exceptionally detailed examination of geological field observations, complemented by satellite data from a relatively small, isolated region. Although the approach and topic of the manuscript seem reasonable, I found that the relevance of the study, beyond that of the local geology, was not sufficiently outlined either in the introductory sections or later in the discussion. This is not to say that the work does not have such implications but that they are not currently described adequately. As such, I think this is probably a moderately easy, yet worthwhile, aspect to resolve as the study clearly has broader implications that would increase the appeal and usefulness to a wider group.

2) Analysis of satellite imagery

The figures showing the satellite data evidently bring a lot to the study in terms of extrapolating the field-based observations to infer more regional processes, and will no doubt be useful for addressing the point outlined above. However, minimal specifics regarding the satellite data (e.g., resolution or age) are provided in the methods section. For example, does all the data presented have the same specifications? In addition, no details are provided regarding the type of analysis or criteria used to interpret features on this data. Related to the latter point, I felt that better use of the satellite data

could have been made by explicitly tying individual features identified in the field-based studies to specific features on the satellite data. If this type of ground truth investigation was undertaken it should be outlined more explicitly in the manuscript. Currently, I think that the lack of the information described in this point partially undermines the findings that are derived from this analysis. As such, I suggest expanding upon these aspects in the relevant sections, but particularly in the methods section.

3) Description of the deformation - fault rock types

The paper adequately describes the orientation and distribution of deformation sufficiently, both in the outcrop observations and also on the satellite data. However, the nature and categorisation of fault rocks could be better described. This is especially important in reactivation studies as the nature of fault rocks is an important line of evidence to evaluate such aspects. As such, I suggest attempting to better categorise the fault rocks, potentially using a scheme such as those outlined in Killick (2003) or Woodcock and Mort (2008).

4) Referencing

The reference list seems up to date and extensive. However, it currently contains three 'submitted' papers, in addition to an 'unpublished' internal report. I appreciate that much of this aspect is beyond the control of the authors. However, I was wondering if it is possible to cite some published work alongside these, perhaps even conference abstracts? For example, the EGU abstract Koehl et al. (2016) appears to address some of the themes in the present study. In addition, if the internal report can be made available online this would be beneficial. Hopefully during the time taken to review and revise the present paper some of the submitted papers will be accepted to alleviate this issue.

In addition to the more substantial points outlined above the following aspects should also be addressed when revising the manuscript:

Abstract – Currently the abstract is quite long and the scientific aims are not easily discernible. Perhaps the abstract can be restricted to the more salient points to assist with this.

Line 13 – 'central Spitsbergen'. For readers not familiar with the geography of Svalbard it might be helpful to say where this is e.g., offshore Northern Norway.

Lines 27-29 – What are the terms in quotes taken from and are they necessary?

Lines 35-39 – The last sentence of the abstract is currently very long. I suggest breaking this into smaller sentences to make it more poignant and easier to follow. This may assist with addressing the point above on the abstract length generally.

Line 36 – 'mildly reactivated'. In my opinion this phrase is ambiguous as it is not clear what would entail 'mild' reactivation compared to an event that could be considered more extensive reactivation. I therefore suggest rewording this in addition to the variants of it that appear throughout the manuscript such as 'partially reactivated' (line 666) and other occurrences (lines 292, 298, 317 and 658). With respect to 'partially reactivated' this is particularly ambiguous as it is unclear whether this is referring to selective structures being reactivated or whether the magnitude of reactivated fault movement is minimal. Please clarify appropriately.

Lines 48-51 – I suggest referring to the location map (Figure 1).

Geological setting – This section is particularly very well written, with the information mostly confined to only the most relevant points, whilst also being generally well organised. However, the authors may want to consider numbering the sections to make this part of the manuscript easier to follow.

Line 145 – This sentence is currently a bit awkward to read. I suggest rewording.

Line 160 – 'Fourth and fifth'. This approach to denoting the points in this paragraph is difficult to follow. I suggest changing it.

Line 166 – Are the phrases in quotes directly from the reference in this sentence? This is currently not clear in the manuscript.

Line 176 – 'thick Pennsylvanian sedimentary strata'. If possible state the thickness of these sediments.

Line 207 – Suggest removing the word 'these' to make the sentence flow better.

Lines 208-209 – It is not clear whether the observations are from this study or those referenced in the sentence. This should be clarified. If both this present study and the previous work make the same observation this should be made clearer.

Line 242 – Consider replacing 'there' with 'here'.

Line 245 – 'the hereby described grey sandstone'. This phrase is quite awkward to read. I suggest rewording.

Line 250 – Suggest removing the word 'rather' to make the sentence flow better.

Lines 267-268 – 'we propose that the hereby-described red-bed sedimentary succession is part of the Hultberget Formation'. The readability of this sentence could be improved. I suggest something like: 'we propose that the red-bed sedimentary succession described herein is part of the Hultberget Formation'.

Line 275 – 'non-cohesive fault-rock'. In line with the second major point outlined above I suggest better characterising this the fault rock.

Line 285 – 'high angle'. If possible, I suggest stating how steep the 'high angle' fault is.

Line 287 – 'cataclasite'. Here, terminology related to fault rocks is used. I suggest doing this elsewhere in the manuscript.

Line 292 – 'during Cenozoic transpression'. When reading the manuscript I did not feel that the evidence leading to this interpretation was adequately provided. Specifically, what is the time constraint leading to this interpretation?

Line 315 – 'is made of'. Consider replacing with 'comprises' to help the sentence flow better.

Lines 349-350 – 'are believed to have been eroded or never deposited'. It is not clear if this is a finding of this study or previous work. I suggest clarifying.

Lines 359-360 – 'which we interpreted as steep brittle faults'. This is an example of the ambiguity outlined in 3rd main point above. In particular, was any attempt made to directly tie the interpretation of the satellite data to actual field observations such as this? If so I suggest stating it more clearly here and elsewhere in the manuscript.

Line 367 – 'dolerite dykes'. Has an age of these dykes been obtained? If so it would be helpful to state it here.

Lines 437-432 – In these opening sentences of the section numerous lines of evidence 'in favour of Mississippian syn-sedimentary extensional brittle faulting' are presented as one very long sentence. It is therefore quite difficult to follow due to the large amount of information contained, and I suggest either numbering the lines of evidence or separating this into multiple sentences.

Line 520 – 'c.' not 'ca.' when not referring to ages or times.

Line 521 – As previous.

Lines 625-665 – The conclusions section contains many long sentences, with each concluding point comprising one such statement. I suggest shortening the sentences to make the conclusions easier to read and more poignant.

Line 631 – Add 's' after 'suggest'.

Line 634 – 'of the Hultberget Formation, thus suggesting'. I suggest breaking this long sentence into two smaller ones by concluding the first after 'Formation' and replacing 'thus' with 'This'. If this is accepted, then 'suggesting' needs to change to 'suggests' in the second statement.

Line 656 – This sentence is incomplete and ends at the word 'which'.

Line 663 – 'gently dipping'. Is it possible to state which way they are dipping?

Line 838 – 'Geochemistry' is spelt incorrectly.

Figure 1A – Scale is missing.

Figure 1B – The white areas on the map are not on the key.

Figure 2 – The green and brown colours on the stratigraphic column do not appear on the key.

Figure 3 – Although the caption states 'The photographs are approximately one kilometer wide' I think more accurate measurement of the scale of the images is required as they are clearly not all the same dimensions.

Figure 4 – The dip markers on the figure are quite problematic to see and on the key a white line in a black box is shown (fault core boundary). However, this does not appear on the figure.

Figure 5 – the field photographs require scale and orientation.

Figure 6 – This figure contains two types of yellow line. Are these showing different features? If so this is not clear in the figure. Also the statement in the caption that 'The outcrop is approximately 10–15 m wide' is a bit ambiguous as it is not entirely clear which parts of the field photo are considered 'outcrop'.

Figure 7A – The lines marked on here are extremely thin and unlikely to be easily visible at publication scale.

Figure 7A-C – Scales needed.

Figure 8 – Scales and orientation need to be provided for all subfigures.

Figure 9 – Same as previous comment.

Figure 10 caption – 'c.' not 'ca.' when not referring to ages or times.

Figure 11 – The text on the figure is very small and unlikely to be easily visible at publication scale.

References

Killick, A.M., 2003, Fault rock classification: An aid to structural interpretation in mine and exploration geology: South African Journal of Geology, v. 106, no. 4, p. 395–402, doi: 10.2113/106.4.395.

Koehl, J., Tveranger, J., Osmundsen, P.T., Braathen, A., Taule, C., and Collombin, M., 2016, Fault-growth deposit in a Carboniferous rift-basin: the Billefjorden Trough , Svalbard: Geophysical Research Abstracts, v. 18, p. 7131.

Perron, P., Guiraud, M., Vennin, E., Moretti, I., Portier, É., Laetitia, L.P., and Konaté, M., 2018, Influence of basement heterogeneity on the architecture of low subsidence rate Paleozoic intracratonic basins (Ahnet and Mouydir basins, Central Sahara): Solid Earth Discussions, doi: 10.5194/se-2018-50.

Phillips, T.B., Jackson, C.A., Bell, R.E., and Duffy, O.B., 2018, Oblique reactivation of lithosphere-scale lineaments controls rift physiography – The upper crustal expression of the Sorgenfrei-Tornquist Zone, offshore southern Norway: Soild Earth, v. 9, p. 403–429, doi: 10.5194/se-9-403-2018.

Woodcock, N.H., and Mort, K.M., 2008, Classification of fault breccias and related fault rocks: Geological Magazine Rapid Communication, v. 145, p. 435–440, doi: 10.1017/S0016756808004883.

---

## Referee Comment (RC2) · A. Lenhart (Referee) · 21 Oct 2018

In their manuscript, Koehl and Munoz-Barrera present new insights on the Carboniferous tectonic and structural evolution of central Spitsbergen. Using observations from satellite images and field evidence from newly exposed outcrops, the authors provide a thorough examination of the local geology and comprehensive discussion of their observations and interpretation. The key results of the study are i) that Mississippian strata was indeed deposited during Mississippian extension, and ii) that Mississippian fault activity and sediment deposition was likely controlled by pre-existing basement-rooted faults and lineaments. As such, the study represents an important contribution

to the understanding of the Paleozoic tectono-stratigraphic history of Svalbard, but also has implications for the understanding of pre-existing structure reactivation. Many recent studies on structural inheritance use relatively low-resolution seismic reflection data, but this study provides actual field evidence for the potential reactivation of basement structures. In general, the manuscript represents a well written and thought-through piece of work. The arguments for the interpretation made are clear and sound. Outcrop photographs are of good quality and support descriptions and interpretations made in the text. My detailed comments outlined below can be grouped into three key recommendations which will further improve the communication of the study results and make the manuscript applicable for a wider audience. If these recommendations are addressed sufficiently, I would like to recommend this manuscript for publication in SED.

Key recommendations:

1) Throughout the manuscript, detailed descriptions of geological structures, formations etc. and the correlations of observations made in Svalbard with similar structures in e.g. the Barents Sea or northern Norway are made. However, in many cases, the location of structures, outcrops etc. is not shown on maps and it is unclear over what distances structural correlations are made. Therefore, some of the correlations and interpretations between structural trends can appear a little farfetched and undermine the good work, especially for readers who are unfamiliar with the geology and tectonic history of the wider study area. Supplementary structural element and plate-tectonic reconstruction maps may help to support the interpretations made by the authors. In general, more references to relevant figures are needed throughout the text.

2) In general, the description of field observations is very detailed and easy to follow. However, I recommend being more quantitative when it comes to extension direction, fault dip, amount of displacement, bed thickness etc. This additional information gives the reader a better idea about the size of structures and enables a better comparison with observations from other field or subsurface studies. In addition, most figures

presented in the manuscript require horizontal and vertical scale bars.

3) The current manuscript is very focused on the reconstruction of the Carboniferous tectonic history of Svalbard, but wider implications of the study results are not discussed. Obvious additional discussion themes could address the role of structure reactivation and stress field perturbations in more detail. Another possibility could be the use of this study as a potential analogue to subsurface studies in the Barents Sea or a comparison of the findings to other studies (e.g. field, subsurface, or modelling studies). Addressing the wider implications of this study will increase the impact of the manuscript and make it applicable to a wider scientific audience.

In addition to these three key recommendations, the following aspects should be addressed in the revision of the manuscript:

Specific comments (text):

Abstract:

The abstract is currently very long and contains complex sentences (e.g. the last two sentences). The rational and motivation of the study is briefly stated in the middle of the abstract (L19-20). However, to emphasize the importance of the study, I suggest moving statements about the study motivation to the first part of the abstract and to add comments on the wider implications of the study. For example: Why is this study important locally and how can the results improve our understanding of e.g. the tectono-stratigraphic evolution of Svalbard? What are the implications for studying basin evolution in the presence of pre-existing basement structures? What is the role of local stress perturbations in fault reactivation? etc.

L24: What is the strike of these basin-oblique, NNE-dipping faults? How do they relate to WNW-ESE-directed extension? Could it be that strike and dip got mixed up and that the faults strike NNE?

L32: pre-existing, not existing

L33: transverse faults, not fault

L37: add commas and write décollements with an é: ...and shallow dipping, bedding parallel, duplex shaped décollements...

L37-38: Out of curiosity – Why would mechanically softer layers such as shales prevent further fault movement? Wouldn't thrust faults preferentially move along the shales? Please clarify your thinking here/in the main text (see later comment L571).

Introduction:

The rational and local importance of the study is well explained in the Introduction. A statement about the wider implications of the study would open it up to a wider audience, provided that a 'wider implications' paragraph is added to the discussion section as well.

L: 70: 'control' would be a better word than 'influence'

Geological Setting:

The geology of the study area is very well described, but the structural elements, formations, and localities that are introduced throughout this section are not shown in Figure 1 (apart from the Billefjorden Fault Zone).

I suggest adding a figure that shows the location and geometry of the geology and structural elements present in the study area in more detail. This will also provide a bit more context and spatial reference to the outcrop photographs shown in later sections of the manuscript. In addition, a regional cross-section across the area may help to illustrate the deformational history and vertical and horizontal relationship between formations better. In general, more references to figures are needed to better guide readers who are unfamiliar with the area. As a suggestion, the authors could include a couple of plate-tectonic reconstructions and structural elements maps in the supplementary material to illustrate the Paleozoic plate configuration of Svalbard, Greenland and Norway, as well as major extensional and compressional events.

L83: Neoproterozoic as one word

L140-141: What was the direction of contraction/plate movement during the Ellesmerian Orogeny? Was it SW?

L145: successions in the footwall and hanging wall of faults?

L168. kilometer-scale

Methods:

The description of the methodology is rather short. The resolution, age, and workflow to interpret the satellite images is not provided.

L201: rephrase; e.g. In areas that are difficult to access, satellite images of exposed basement rocks were used to identify brittle faults in exposed Proterozoic basement rocks. . ..

Results:

Basement rocks: L219-221: Can you indicate the faults that cross-cut the Atomfjella on a map? Where is Ny-Friesland?

L224-225: See previous comment. Please indicate the mentioned localities on a map, otherwise the reader has no idea about the location and distance between areas with WNW-ESE-trending faults and basement structures. A map will help to support your interpretation.

Sedimentary rocks: L234: south-to-southwestward

L241: How thin are these beds? Be quantitative.

L244: '. . .previous descriptions. Plural.

L248: Remove 'However'. Start sentence with: Iron nodules found in the upper part. . .

L250: Replace 'On the contrary' with 'However',

L259 and throughout this paragraph: How thick are the described sandstone and shale beds? There is no scale in the photograph in Figure 7.

Brittle faults: L276: You state the amount of displacement along these faults in the figure caption, can you also add it in the text?

L278: Can you quantify the amount of thickening?

L288: décollements

L292: décollements

L307: cross-cut

L301 & 303: cross-cutting

L315: cross-cut

L320: Is it possible to estimate the amount of displacement across the Overgangshytta Fault? e.g. order of magnitude. I see that you provided an estimate on L355, but it would be nice to also have this in the results section.

Discussion:

The discussion section represents a very thorough examination and discussion of possible interpretations for the observed structures. Parts of the discussion/interpretation can be supported by additional figures to support the author's arguments and to better guide the reader. The current manuscript does not include a section on the wider implications of the results of this study. I suggest to add a paragraph on this at the end of the discussion section.

L325: The first sentence of the Discussion section repeats the last sentence of the previous paragraph (L318-321). I suggest rephrasing these sentences to avoid too much repetition.

L328-330: This sentence suggests that, based on the fault core width and amount of

deformation, the Overgangshytta Fault does not terminate nearby. Can you support this interpretation with a reference to studies that investigated the relationship between fault length/displacement and deformation zone size?

L345: kilometer-thick

L348: meter-to-kilometer-scale, down-to-NNE

L360-381: It is difficult to believe how basement structures in Spitsbergen correlate to fault zones in northern Norway without showing plate-tectonic reconstructions (see earlier comments on the lack of supporting figures). The Timanian Orogeny has not been introduced at the beginning of the manuscript. At the moment, the interpretation of the WNW-ESE-striking faults appears to be based on long-distance, map-view correlations and may seem a little farfetched. However, additional figures illustrating the geometrical and plate-tectonic relationship between the correlates basement structures in Spitsbergen, the Barents Sea, and northern Norway may support and clarify the presented interpretation.

L410-411: Can you quantify the amount of reverse displacement along the fault? e.g. meter-scale or tens-of-meter?

L412: décollement

L416: What is the scale of these 'minor thrust faults'?

L428: décollements

L435 and following paragraph: What is the dominant extension direction during the Mississippian? How does it relate to the N-S, NE-SW, and WNW-ESE-striking faults observed in the area? Was there a preferential reactivation of faults oriented perpendicular to the extension direction? Or may local strain perturbations be responsible for the activation of basin-oblique faults?

L439: Can you quantify the amount of thickening? It looks very minor on the outcrop

photograph in Figure 8. Please add vertical and horizontal scales to every figure.

L440: cross-cutting

L447: is believed

L450: paleo-current data

L465-469: This sentence is very long and complex. Please rephrase. Add commas between shallow-dipping, bedding-parallel, duplex-shaped décollements.

L470-476: See previous comment above. It is difficult to picture the spatial and geometrical relationship between WNW-ESE-striking faults in Spitsbergen, northern Norway and Greenland without any maps. These seem to be very long-distance correlations unless you show that these faults originate from the same locality during Late Devonian-Carboniferous.

L491: Again, what is the Mississippian extension direction? How does the stress field look like?

L493: cross-cutting

L497: Please quantify the dip angle of the Billefjorden Group

L508: (b) not (a)

L512: Where is Kongsfjorden and the Brøggerhalvøya located? Please indicate on a map.

L523: local absence of the Late Mississippian unconformity

L533: What is the direction of compression/transpression?

L540: How far away is the Finnmark Platform from the study area? This seems to be a very long/distance correlation.

L546 and following paragraph: What was the extension direction? Was it stable or

did it change? Can the activity of faults that are not preferentially aligned towards the extension direction be explained by local, potentially basement fabric-controlled, stress/strain perturbations? It would be nice to illustrate fault activity (e.g. initiation phase, interaction and linkage phase etc.) and extension direction through time on map-view sketches.

L571: décollements; How thick are the shale beds? Are they thick enough to decouple faulting on N-S faults from WNW-ESE faults? It would be good to support this statement with a literature reference, e.g. studies on mechanical stratigraphy (Wilkins, S. J., & Gross, M. R. (2002). Normal fault growth in layered rocks at Split Mountain, Utah: influence of mechanical stratigraphy on dip linkage, fault restriction and fault scaling. Journal of Structural Geology, 24(9), 1413-1429.)

L577: cross-cut

L578: Please quantify the amount of offset

L582: small amounts: plural

Conclusions:

Each conclusion point consists of a single, very long and complex sentence. Please consider breaking them up into multiple sentences to make it easier to follow them. Consider adding a conclusion point that illustrates the wider implications of your study results.

L650: pre-existing Neoproterozoic faults; remove 'which' at the end of the sentence.

L663: décollements

L666: décollements

Specific comments (figures):

Figure 1B: The map doesn't show many localities and formations that are mentioned

[Figure]

in the text. Please add them. It would also be useful to have a structural elements map for the Late Devonian-Carboniferous covering Svalbard, the Barents Sea, northern Norway and Greenland (see comments above). A map like this would make it easier to follow your thinking and interpretations.

Figure 2: The orange and green colours shown in the stratigraphic chart are not explained in the legend. Please add them.

Figure 3: Although you stated an approximate scale of each satellite image at the end of the figure caption, please add a scale bar in every image. The interpreted foliation and lineaments are actually difficult to see on the dark rocks. Is there any change to improve the image quality?

Figure 4: What do the pink and blue arrows indicate? Not all brittle faults have a dip direction indicator? Is the dip of these faults unknown?

Figure 5: Please add vertical and horizontal scales to photograph A. The label 'Fig. 4b' in photograph A seems to be wrong.

Figure 6: Please add horizontal and particularly, vertical scale bars. An approximate outcrop size is not enough.

Figure 7: Please add horizontal and vertical scale bars. Location of 7A is not indicated in Figure 4.

Figure 8: Please add horizontal and particularly, vertical scale bars – at least in B and C. An approximate outcrop size is not enough. Indicate the location of these outcrops on Figure 4.

Figure 9: Please add horizontal and particularly, vertical scale bars. An approximate outcrop size is not enough. Indicate the location of these outcrops on Figure 4.

Figure 10: Please add horizontal and particularly, vertical scale bars. An approximate outcrop size is not enough. Indicate the location of these outcrops on Figure 4. Add

'southeastward view of the Overgangshytta Fault' for the description of A in Figure caption. Location of 10D is not shown in 10A.

Figure 11: Please indicate profile location in Figure 4 and add approximate horizontal and vertical scales. Profiles like this greatly help the reader to follow the description of your observations and interpretations. It might be useful to refer to this figure earlier in the manuscript, e.g. in the results section.

Figure captions: - Replace crosscut with cross-cut where applicable

---

## Author Comment (AC1) · 27 Nov 2018

Dear Dr. Peace, Thank you very much for your input on the manuscript, it is highly appreciated. Here is our response to your comments. We hope the changes we implemented improve the shortcomings of the manuscript highlighted by your comments and suggestions. Please do not hesitate to contact us shall this not be the case for some comments.

1. Comments from Dr. Peace Comment 1: Wider implications of the study and comparison to other regions The authors present an exceptionally detailed examination of geological field observations, complemented by satellite data from a relatively small,

isolated region. Although the approach and topic of the manuscript seem reasonable, I found that the relevance of the study, beyond that of the local geology, was not sufficiently outlined either in the introductory sections or later in the discussion. This is not to say that the work does not have such implications but that they are not currently described adequately. As such, I think this is probably a moderately easy, yet worthwhile, aspect to resolve as the study clearly has broader implications that would increase the appeal and usefulness to a wider group. Comment 2: Analysis of satellite imagery The figures showing the satellite data evidently bring a lot to the study in terms of extrapolating the field-based observations to infer more regional processes, and will no doubt be useful for addressing the point outlined above. However, minimal specifics regarding the satellite data (e.g., resolution or age) are provided in the methods section. For example, does all the data presented have the same specifications? In addition, no details are provided regarding the type of analysis or criteria used to interpret features on this data. Related to the latter point, I felt that better use of the satellite data could have been made by explicitly tying individual features identified in the field-based studies to specific features on the satellite data. If this type of ground truth investigation was undertaken it should be outlined more explicitly in the manuscript. Currently, I think that the lack of the information described in this point partially undermines the findings that are derived from this analysis. As such, I suggest expanding upon these aspects in the relevant sections, but particularly in the methods section. Comment 3: Description of the deformation - fault rock types The paper adequately describes the orientation and distribution of deformation sufficiently, both in the outcrop observations and also on the satellite data. However, the nature and categorisation of fault rocks could be better described. This is especially important in reactivation studies as the nature of fault rocks is an important line of evidence to evaluate such aspects. As such, I suggest attempting to better categorise the fault rocks, potentially using a scheme such as those outlined in Killick (2003) or Woodcock and Mort (2008). Comment 4: Referencing The reference list seems up to date and extensive. However, it currently contains three 'submitted' papers, in addition to an 'unpublished' internal report. I appreciate that much of

this aspect is beyond the control of the authors. However, I was wondering if it is possible to cite some published work alongside these, perhaps even conference abstracts? For example, the EGU abstract Koehl et al. (2016) appears to address some of the themes in the present study. In addition, if the internal report can be made available online this would be beneficial. Hopefully during the time taken to review and revise the present paper some of the submitted papers will be accepted to alleviate this issue. Comment 5: Abstract – Currently the abstract is quite long and the scientific aims are not easily discernible. Perhaps the abstract can be restricted to the more salient points to assist with this. Comment 6: Line 13 – 'central Spitsbergen'. For readers not familiar with the geography of Svalbard it might be helpful to say where this is e.g., offshore Northern Norway. Comment 7: Lines 27-29 – What are the terms in quotes taken from and are they necessary? Comment 8: Lines 35-39 – The last sentence of the abstract is currently very long. I suggest breaking this into smaller sentences to make it more poignant and easier to follow. This may assist with addressing the point above on the abstract length generally. Comment 9: Line 36 – 'mildly reactivated'. In my opinion this phrase is ambiguous as it is not clear what would entail 'mild' reactivation compared to an event that could be considered more extensive reactivation. I therefore suggest rewording this in addition to the variants of it that appear throughout the manuscript such as 'partially reactivated' (line 666) and other occurrences (lines 292, 298, 317 and 658). With respect to 'partially reactivated' this is particularly ambiguous as it is unclear whether this is referring to selective structures being reactivated or whether the magnitude of reactivated fault movement is minimal. Please clarify appropriately. Comment 10: Lines 48-51 – I suggest referring to the location map (Figure 1). Comment 11: Geological setting – This section is particularly very well written, with the information mostly confined to only the most relevant points, whilst also being generally well organised. However, the authors may want to consider numbering the sections to make this part of the manuscript easier to follow. Comment 12: Line 145 – This sentence is currently a bit awkward to read. I suggest rewording. Comment 13: Line 160 – 'Fourth and fifth'. This approach to denoting the points in this paragraph is difficult to follow. I suggest changing it. Comment 14: Line 166 – Are the phrases in quotes directly from the reference in this sentence? This is currently not clear in the manuscript. Comment 15: Line 176 – 'thick Pennsylvanian sedimentary strata'. If possible state the thickness of these sediments. Comment 16: Line 207 – Suggest removing the word 'these' to make the sentence flow better. Comment 17: Lines 208-209 – It is not clear whether the observations are from this study or those referenced in the sentence. This should be clarified. If both this present study and the previous work make the same observation this should be made clearer. Comment 18: Line 242 – Consider replacing 'there' with 'here'. Comment 19: Line 245 – 'the hereby described grey sandstone'. This phrase is quite awkward to read. I suggest rewording. Comment 20: Line 250 – Suggest removing the word 'rather' to make the sentence flow better. Comment 21: Lines 267-268 – 'we propose that the hereby-described red-bed sedimentary succession is part of the Hultberget Formation'. The readability of this sentence could be improved. I suggest something like: 'we propose that the red-bed sedimentary succession described herein is part of the Hultberget Formation'. Comment 22: Line 275 – 'non-cohesive fault-rock'. In line with the second major point outlined above I suggest better characterising this the fault rock. Comment 23: Line 285 – 'high angle'. If possible, I suggest stating how steep the 'high angle' fault is. Comment 24: Line 287 – 'cataclasite'. Here, terminology related to fault rocks is used. I suggest doing this elsewhere in the manuscript. Comment 25: Line 292 – 'during Cenozoic transpression'. When reading the manuscript I did not feel that the evidence leading to this interpretation was adequately provided. Specifically, what is the time constraint leading to this interpretation? Comment 26: Line 315 – 'is made of'. Consider replacing with 'comprises' to help the sentence flow better. Comment 27: Lines 349-350 – 'are believed to have been eroded or never deposited'. It is not clear if this is a finding of this study or previous work. I suggest clarifying. Comment 28: Lines 359-360 – 'which we interpreted as steep brittle faults'. This is an example of the ambiguity outlined in 3rd main point above. In particular, was any attempt made to directly tie the interpretation of the satellite data to actual field observations such as this? If so I suggest stating it more clearly here and elsewhere in

the manuscript. Comment 29: Line 367 – 'dolerite dykes'. Has an age of these dykes been obtained? If so it would be helpful to state it here. Comment 30: Lines 437-432 – In these opening sentences of the section numerous lines of evidence 'in favour of Mississippian syn-sedimentary extensional brittle faulting' are presented as one very long sentence. It is therefore quite difficult to follow due to the large amount of information contained, and I suggest either numbering the lines of evidence or separating this into multiple sentences. Comment 31: Line 520 – 'c.' not 'ca.' when not referring to ages or times. Comment 32: Line 521 – As previous. Comment 33: Lines 625-665 – The conclusions section contains many long sentences, with each concluding point comprising one such statement. I suggest shortening the sentences to make the conclusions easier to read and more poignant. Comment 34: Line 631 – Add 's' after 'suggest'. Comment 35: Line 634 – 'of the Hultberget Formation, thus suggesting'. I suggest breaking this long sentence into two smaller ones by concluding the first after 'Formation' and replacing 'thus' with 'This'. If this is accepted, then 'suggesting' needs to change to 'suggests' in the second statement. Comment 36: Line 656 – This sentence is incomplete and ends at the word 'which'. Comment 37: Line 663 – 'gently dipping'. Is it possible to state which way they are dipping? Comment 38: Line 838 – 'Geochemistry' is spelt incorrectly. Comment 39: Figure 1A – Scale is missing. Comment 40: Figure 1B – The white areas on the map are not on the key. Comment 41: Figure 2 – The green and brown colours on the stratigraphic column do not appear on the key. Comment 42: Figure 3 – Although the caption states 'The photographs are approximately one kilometer wide' I think more accurate measurement of the scale of the images is required as they are clearly not all the same dimensions. Comment 43: Figure 4 – The dip markers on the figure are quite problematic to see and on the key a white line in a black box is shown (fault core boundary). However, this does not appear on the figure. Comment 44: Figure 5 – the field photographs require scale and orientation. Comment 45: Figure 6 – This figure contains two types of yellow line. Are these showing different features? If so this is not clear in the figure. Also the statement in the caption that 'The outcrop is approximately 10–15 m wide' is a bit ambiguous as

it is not entirely clear which parts of the field photo are considered 'outcrop'. Comment 46: Figure 7A – The lines marked on here are extremely thin and unlikely to be easily visible at publication scale. Comment 47: Figure 7A-C – Scales needed. Comment 48: Figure 8 – Scales and orientation need to be provided for all subfigures. Comment 49: Figure 9 – Same as previous comment. Comment 50: Figure 10 caption – 'c.' not 'ca.' when not referring to ages or times. Comment 51: Figure 11 – The text on the figure is very small and unlikely to be easily visible at publication scale. Comment 52: References Killick, A.M., 2003, Fault rock classification: An aid to structural interpretation in mine and exploration geology: South African Journal of Geology, v. 106, no. 4, p. 395–402, doi: 10.2113/106.4.395. Koehl, J., Tveranger, J., Osmundsen, P.T., Braathen, A., Taule, C., and Collombin, M., 2016, Fault-growth deposit in a Carboniferous rift-basin: the Billefjorden Trough , Svalbard: Geophysical Research Abstracts, v. 18, p. 7131. Perron, P., Guiraud, M., Vennin, E., Moretti, I., Portier, É., Laetitia, L.P., and Konaté, M., 2018, Influence of basement heterogeneity on the architecture of low subsidence rate Paleozoic intracratonic basins (Ahnet and Mouydir basins, Central Sahara): Solid Earth Discussions, doi: 10.5194/se-2018-50. Phillips, T.B., Jackson, C.A., Bell, R.E., and Duffy, O.B., 2018, Oblique reactivation of lithosphere-scale lineaments controls rift physiography – The upper crustal expression of the Sorgenfrei-Tornquist Zone, offshore southern Norway: Soild Earth, v. 9, p. 403–429, doi: 10.5194/se-9-403-2018. Woodcock, N.H., and Mort, K.M., 2008, Classification of fault breccias and related fault rocks: Geological Magazine Rapid Communication, v. 145, p. 435–440, doi:10.1017/S0016756808004883.

2. Author's response Comment 1: agreed. Comment 2: agreed. Comment 3: agreed. We now use the classification of Woodcock and Mort (2008). Comment 4: agreed. The internal report is already available on the main author's ResearchGate webpage upon request. However, the suggested abstract by Koehl et al. (2016) does complement any of the submitted papers. Comment 5: agreed. Comment 6: agreed. However, "offshore northern Norway" is quite confusing for readers that are actually familiar with the study area. Comment 7: agreed, they are not necessary in the abstract and can

be described at a later stage, in the discussion. Comment 8: agreed. Comment 9: disagreed. The term "mildly" refers to the magnitude of movement along the reactivated structures, which is relatively small compared to km-scale offsets along large faults (e.g., the Billefjorden Fault Zone) in the study area. Furthermore, the term is clarified line 441 where it is followed by "with little or no upwards propagation". Comment 10: agreed. Comment 11: agreed. Comment 12: agreed. Comment 13: disagreed. The introductory sentence of the paragraph stipulates that the paragraph is dealing with five different formations, and we therefore believe that the use of "first", "second", etc. appropriate to this paragraph. Comment 14: yes, the phrases in between quotation marks are directly from the associated publication. The manuscript even specify in which figure of the referred publication one may find the terms in quotation marks: "Gawthorpe and Leeder, 2000, their fig. 3". Comment 15: agreed. Comment 16: agreed. Comment 17: the first sentence lines 207–208 refers to literature data, while the second sentence shows that the gneissic foliation described in the literature can be observed on satellite images. Comment 18: agreed. Comment 19: agreed. Comment 20: agreed. Comment 21: agreed. Comment 22: agreed. Comment 23: agreed. Comment 24: agreed. Comment 25: there is no major post-Mississippian contractional–transpressional tectonic event recorded in Spitsbergen other than an episode of Cenozoic transpression. Thus, it is natural to infer that any contractional structure or reactivation might have formed during Cenozoic transpression. Comment 26: agreed. Comment 27: agreed. Comment 28: agreed. Comment 29: agreed. Comment 30: agreed. Comment 31: agreed. However, the examples lines 520 and 521 should remain as "ca." since they are referring to ages. Comment 32: see response to comment 31. Comment 33: disagreed. The present manuscript addresses a very specific issue (initiation of extension in Mississippian times, not in Early Pennsylvanian) and the authors need to be very specific in their conclusions in order to make their findings clear for all specialists and maximize the impact of the paper on future research. Comment 34: disagreed. Two arguments "suggest" this: the extensional growth strata and the change of contact type between the two formations. Comment 35: agreed. Comment 36: agreed. Comment

37: the dip of the décollements varies as that of Carboniferous strata in the area, i.e., from SW to SE and from NW to NE. The authors believe that this information is not relevant to include to the conclusion and would rather overload a conclusion already crowded with specific points. Comment 38: agreed. Comment 39: agreed. Comment 40: agreed. Comment 41: agreed. Comment 42: agreed. Comment 43: agreed. Comment 44: disagreed. All four figures in figure 5 already contain scales and do not need orientation since they do not show oriented structures. Comment 45: agreed. However, the distinction between dotted and dashed yellow lines is made in the figure caption. Comment 46: agreed. Comment 47: agreed. Comment 48: agreed. Comment 49: agreed. Comment 50: agreed. Comment 51: agreed. Comment 52: agreed. However, the authors do not understand the suggestion of the work by Phillips et al. (2018) and Perron et al. (2018) to the reference list, although the authors are familiar with the suggested works. Perhaps the referee could specify the aim and the place he may find appropriate to add these references.

3. Changes implemented Comment 1: addition of a sentence on the broader implications of the present study on the hydrocarbon exploration, geodynamics, and margin architecture at the end of paragraph 1 in the introduction: "The present local study has broader regional implications, especially regarding the geodynamic setting of Arctic regions in the Mississippian (contraction versus extension versus tectonic quiescence?), the architecture and geometry of the Barents Sea and west Spitsbergen margins (Mississippian basins?), and may affect our understanding of the distribution of Mississippian coal-bearing hydrocarbon source rock in the Barents Sea" lines 60–64. Comment 2: addition of "In addition, fault surfaces and escarpments in the field were tied to map-view lineaments on satellite images that matched their trend and location (Figure 4). Critical factors used in the interpretation of geological features on satellite images in inaccessible areas include existing literature (e.g., N–S-trending gneissic foliation in basement rocks east and southeast of the field area was evidenced by multiple works, including notably Harland et al., 1966 and Witt-Nilsson et al., 1998), the geological database at svalbardkartet.npolar.no, and similarities with fault-related escarpments

tied to actual brittle faults in the field area (Figure 4). Glacial features were segregated from ductile and brittle structures and fabrics using satellite images and scientific literature on recent and past glacial flow. Satellite images used in the present study are from 2011 and have a horizontal resolution of 40 cm" to the method chapter. Comment 3: addition of "fine-grained," line 275; "i.e., fault gouge; Woodcock and Mort, 2008;" line 275; "dominantly fine-grained cohesive fault-rock (i.e., meso- to ultra-cataclasite; Woodcock and Mort, 2008)" lines 287–288; "(meso- to ultra-)" line 1064; Reference to Woodcock and Mort (2008) to the reference list. Comment 4: Added Reference to Bergh et al. (2014), Koehl (2018), and Klitzke et al. (2018) as complements to Bergh et al. (submitted), Koehl et al. (submitted), and Klitzke et al. (submitted) respectively. Comment 5: deletion of one sentence and several phrases. Comment 6: added "Svalbard" line 14. Comment 7: deleted terms in quotation marks. Comment 8: the sentence was shortened. Comment 9: no change. Comment 10: added "figure 1" line 49. Comment 11: added numbering to Geological setting sub-chapters. Comment 12: sentence split into two and partially rewritten. Comment 13: no change. Comment 14: no change. Comment 15: added "tens (hundreds?) of meters" line 181. Comment 16: implemented suggested change. Comment 17: no change. Comment 18: replaced "there" by "at this location". Comment 19: deleted "hereby described". Comment 20: deleted "rather". Comment 21: implemented suggested change. Comment 22: implemented suggested change. See answer to comment 3. Comment 23: added "(> 70°)". Comment 24: implemented suggested change. See answer to comment 3. Comment 25: no change. Comment 26: implemented suggested change. Comment 27: added reference to Harland et al. (1974). Comment 28: see response to comment 2. Also added "based on their similarities with fault-related lineaments in the field area (Figure 4) and their obliquity to the dominant N–S-trending ductile fabrics and structures (Harland et al., 1966; Balashov et al., 1993; Witt-Nilsson et al., 1998; Johansson and Gee, 1999)" lines 377–379. Comment 29: deletion of "in Mississippian times (Visean; Lippard and Prestvik, 1997)" line 386, and addition of "Mississippian (Visean; Lippard and Prestvik, 1997)" line 387. Comment 30: addition of numbers ahead of each evidence. Comment 31: replaced "ca." by "approximately" where needed in main text. Comment 32: none. Comment 33: none. Comment 34: none. Comment 35: implemented suggested changes. Comment 36: deletion of "which" and addition of ".". Comment 37: none. Comment 38: implemented suggested change. Comment 39: implemented suggested change. Comment 40: addition of "Areas shaded in white represent glaciers" to the figure caption. Comment 41: implemented suggested change. Comment 42: addition of a common scale on figure 3a. Comment 43: implemented suggested change. Comment 44: none. Comment 45: added scale to the figure. Comment 46: implemented suggested change. Comment 47: implemented suggested changes. Comment 48: implemented suggested changes. Comment 49: implemented suggested changes. Comment 50: implemented suggested change. Comment 51: increased the size of all text in the figure. Comment 52: addition of the Woodcock and Mort (2008) reference to the reference list.

---

## Author Comment (AC2) · 27 Nov 2018

Dear Dr. Lenhart, thank you very much for your input on the manuscript, it is highly appreciated. Here is our response to your comments. We hope the changes we implemented improve the shortcomings of the manuscript highlighted by your comments and suggestions. Please do not hesitate to contact us shall this not be the case for some comments.

1. Comments from Dr. Lenhart Comment 1: Throughout the manuscript, detailed descriptions of geological structures, formations etc. and the correlations of observations made in Svalbard with similar structures in e.g. the Barents Sea or northern Norway

are made. However, in many cases, the location of structures, outcrops etc. is not shown on maps and it is unclear over what distances structural correlations are made. Therefore, some of the correlations and interpretations between structural trends can appear a little farfetched and undermine the good work, especially for readers who are unfamiliar with the geology and tectonic history of the wider study area. Supplementary structural element and plate-tectonic reconstruction maps may help to support the interpretations made by the authors. In general, more references to relevant figures are needed throughout the text.

Comment 2: In general, the description of field observations is very detailed and easy to follow. However, I recommend being more quantitative when it comes to extension direction, fault dip, amount of displacement, bed thickness etc. This additional information gives the reader a better idea about the size of structures and enables a better comparison with observations from other field or subsurface studies. In addition, most figures presented in the manuscript require horizontal and vertical scale bars.

Comment 3: The current manuscript is very focused on the reconstruction of the Carboniferous tectonic history of Svalbard, but wider implications of the study results are not discussed. Obvious additional discussion themes could address the role of structure reactivation and stress field perturbations in more detail. Another possibility could be the use of this study as a potential analogue to subsurface studies in the Barents Sea or a comparison of the findings to other studies (e.g. field, subsurface, or modelling studies). Addressing the wider implications of this study will increase the impact of the manuscript and make it applicable to a wider scientific audience.

Comment 4: The abstract is currently very long and contains complex sentences (e.g. the last two sentences). The rational and motivation of the study is briefly stated in the middle of the abstract (L19-20). However, to emphasize the importance of the study, I suggest moving statements about the study motivation to the first part of the abstract and to add comments on the wider implications of the study. For example: Why is this study important locally and how can the results improve our understanding of e.g.

the tectono-stratigraphic evolution of Svalbard? What are the implications for studying basin evolution in the presence of pre-existing basement structures? What is the role of local stress perturbations in fault reactivation? etc. Comment 5: L24: What is the strike of these basin-oblique, NNE-dipping faults? How do they relate to WNW-ESE-directed extension? Could it be that strike and dip got mixed up and that the faults strike NNE?

Comment 6: L32: pre-existing, not existing

Comment 7: L33: transverse faults, not fault

Comment 8: L37: add commas and write décollements with an é: : : :and shallow dipping, bedding parallel, duplex shaped décollements: : :

Comment 9: L37-38: Out of curiosity – Why would mechanically softer layers such as shales prevent further fault movement? Wouldn't thrust faults preferentially move along the shales? Please clarify your thinking here/in the main text (see later comment L571).

Comment 10: Introduction: The rational and local importance of the study is well explained in the Introduction. A statement about the wider implications of the study would open it up to a wider audience, provided that a 'wider implications' paragraph is added to the discussion section as well.

Comment 11: L: 70: 'control' would be a better word than 'influence'

Comment 12: Geological Setting: The geology of the study area is very well described, but the structural elements, formations, and localities that are introduced throughout this section are not shown in Figure 1 (apart from the Billefjorden Fault Zone).

Comment 13: I suggest adding a figure that shows the location and geometry of the geology and structural elements present in the study area in more detail. This will also provide a bit more context and spatial reference to the outcrop photographs shown in later sections of the manuscript. In addition, a regional cross-section across the area may help to illustrate the deformational history and vertical and horizontal relationship

between formations better.

Comment 14: In general, more references to figures are needed to better guide readers who are unfamiliar with the area. As a suggestion, the authors could include a couple of plate-tectonic reconstructions and structural elements maps in the supplementary material to illustrate the Paleozoic plate configuration of Svalbard, Greenland and Norway, as well as major extensional and compressional events.

Comment 15: L83: Neoproterozoic as one word

Comment 16: L140-141: What was the direction of contraction/plate movement during the Ellesmerian Orogeny? Was it SW?

Comment 17: L145: successions in the footwall and hanging wall of faults?

Comment 18: L168. kilometer-scale

Comment 19: Methods: The description of the methodology is rather short. The resolution, age, and workflow to interpret the satellite images is not provided.

Comment 20: L201: rephrase; e.g. In areas that are difficult to access, satellite images of exposed basement rocks were used to identify brittle faults in exposed Proterozoic basement rocks: : :.

Comment 21: Results: Basement rocks: L219-221: Can you indicate the faults that cross-cut the Atomfjella on a map? Where is Ny-Friesland?

Comment 22: L224-225: See previous comment. Please indicate the mentioned localities on a map, otherwise the reader has no idea about the location and distance between areas with WNW-ESE-trending faults and basement structures. A map will help to support your interpretation.

Comment 23: Sedimentary rocks: L234: south-to-southwestward

Comment 24: L241: How thin are these beds? Be quantitative.

[Figure]

Comment 25: L244: ': : :previous descriptions. Plural.

Comment 26: L248: Remove 'However'. Start sentence with: Iron nodules found in the upper part: : :

Comment 27: L250: Replace 'On the contrary' with 'However',

Comment 28: L259 and throughout this paragraph: How thick are the described sandstone and shale beds? There is no scale in the photograph in Figure 7.

Comment 29: Brittle faults: L276: You state the amount of displacement along these faults in the figure caption, can you also add it in the text?

Comment 30: L278: Can you quantify the amount of thickening?

Comment 31: L288: décollements

Comment 32: L292: décollements

Comment 33: L307: cross-cut

Comment 34: L301 & 303: cross-cutting

Comment 35: L315: cross-cut

Comment 36: L320: Is it possible to estimate the amount of displacement across the Overgangshytta Fault? e.g. order of magnitude. I see that you provided an estimate on L355, but it would be nice to also have this in the results section.

Comment 37: Discussion: The discussion section represents a very thorough examination and discussion of possible interpretations for the observed structures. Parts of the discussion/interpretation can be supported by additional figures to support the author's arguments and to better guide the reader. The current manuscript does not include a section on the wider implications of the results of this study. I suggest to add a paragraph on this at the end of the discussion section.

Comment 38: L325: The first sentence of the Discussion section repeats the last sentence of the previous paragraph (L318-321). I suggest rephrasing these sentences to avoid too much repetition.

Comment 39: L328-330: This sentence suggests that, based on the fault core width and amount of deformation, the Overgangshytta Fault does not terminate nearby. Can you support this interpretation with a reference to studies that investigated the relationship between fault length/displacement and deformation zone size?

Comment 40: L345: kilometer-thick

Comment 41: L348: meter-to-kilometer-scale, down-to-NNE

Comment 42: L360-381: It is difficult to believe how basement structures in Spitsbergen correlate to fault zones in northern Norway without showing plate-tectonic reconstructions (see earlier comments on the lack of supporting figures). The Timanian Orogeny has not been introduced at the beginning of the manuscript. At the moment, the interpretation of the WNW-ESE-striking faults appears to be based on long-distance, map-view correlations and may seem a little farfetched. However, additional figures illustrating the geometrical and plate-tectonic relationship between the correlates basement structures in Spitsbergen, the Barents Sea, and northern Norway may support and clarify the presented interpretation.

Comment 43: L410-411: Can you quantify the amount of reverse displacement along the fault? e.g. meter-scale or tens-of-meter?

Comment 44: L412: décollement

Comment 45: L416: What is the scale of these 'minor thrust faults'?

Comment 46: scale of these 'minor thrust faults'?

Comment 47: L435 and following paragraph: What is the dominant extension direction during the Mississippian? How does it relate to the N-S, NE-SW, and WNW-ESE-striking faults observed in the area? Was there a preferential reactivation of faults

oriented perpendicular to the extension direction? Or may local strain perturbations be responsible for the activation of basin-oblique faults?

Comment 48: L439: Can you quantify the amount of thickening? It looks very minor on the outcrop photograph in Figure 8. Please add vertical and horizontal scales to every figure.

Comment 49: L440: cross-cutting

Comment 50: L447: is believed

Comment 51: L450: paleo-current data

Comment 52: L465-469: This sentence is very long and complex. Please rephrase. Add commas between shallow-dipping, bedding-parallel, duplex-shaped décollements.

Comment 53: L470-476: See previous comment above. It is difficult to picture the spatial and geometrical relationship between WNW-ESE-striking faults in Spitsbergen, northern Norway and Greenland without any maps. These seem to be very long-distance correlations unless you show that these faults originate from the same locality during Late Devonian-Carboniferous.

Comment 54: L491: Again, what is the Mississippian extension direction? How does the stress field look like?

Comment 55: L493: cross-cutting

Comment 56: L497: Please quantify the dip angle of the Billefjorden Group

Comment 57: L508: (b) not (a)

Comment 58: L512: Where is Kongsfjorden and the Brøggerhalvøya located? Please indicate on a map.

Comment 59: L523: local absence of the Late Mississippian unconformity

Comment 60: L533: What is the direction of compression/transpression?

Comment 61: L540: How far away is the Finnmark Platform from the study area? This seems to be a very long/distance correlation.

Comment 62: L546 and following paragraph: What was the extension direction? Was it stable or did it change? Can the activity of faults that are not preferentially aligned towards the extension direction be explained by local, potentially basement fabric-controlled, stress/strain perturbations? It would be nice to illustrate fault activity (e.g. initiation phase, interaction and linkage phase etc.) and extension direction through time on map-view sketches.

Comment 63: L571: décollements; How thick are the shale beds? Are they thick enough to decouple faulting on N-S faults fromWNW-ESE faults? It would be good to support this statement with a literature reference, e.g. studies on mechanical stratigraphy (Wilkins, S. J., & Gross, M. R. (2002). Normal fault growth in layered rocks at Split Mountain, Utah: influence of mechanical stratigraphy on dip linkage, fault restriction and fault scaling. Journal of Structural Geology, 24(9), 1413-1429.)

Comment 64: L577: cross-cut

Comment 65: L578: Please quantify the amount of offset

Comment 66: L582: small amounts: plural

Comment 67: Conclusions: Each conclusion point consists of a single, very long and complex sentence. Please consider breaking them up into multiple sentences to make it easier to follow them. Consider adding a conclusion point that illustrates the wider implications of your study results.

Comment 68: L650: pre-existing Neoproterozoic faults; remove 'which' at the end of the sentence.

Comment 69: L663: décollements

Comment 70: L666: décollements

Comment 71: Figure 1B: The map doesn't show many localities and formations that are mentioned in the text. Please add them. It would also be useful to have a structural elements map for the Late Devonian-Carboniferous covering Svalbard, the Barents Sea, northern Norway and Greenland (see comments above). A map like this would make it easier to follow your thinking and interpretations.

Comment 72: Figure 2: The orange and green colours shown in the stratigraphic chart are not explained in the legend. Please add them.

Comment 73: Figure 3: Although you stated an approximate scale of each satellite image at the end of the figure caption, please add a scale bar in every image. The interpreted foliation and lineaments are actually difficult to see on the dark rocks. Is there any change to improve the image quality?

Comment 74: Figure 4: What do the pink and blue arrows indicate? Not all brittle faults have a dip direction indicator? Is the dip of these faults unknown?

Comment 75: Figure 5: Please add vertical and horizontal scales to photograph A. The label 'Fig. 4b' in photograph A seems to be wrong. Comment 76: Figure 6: Please add horizontal and particularly, vertical scale bars. An approximate outcrop size is not enough.

Comment 77: Figure 7: Please add horizontal and vertical scale bars. Location of 7A is not indicated in Figure 4.

Comment 78: Figure 8: Please add horizontal and particularly, vertical scale bars – at least in B and C. An approximate outcrop size is not enough. Indicate the location of these outcrops on Figure 4.

Comment 79: Figure 9: Please add horizontal and particularly, vertical scale bars. An approximate outcrop size is not enough. Indicate the location of these outcrops on Figure 4.

Comment 80: Figure 10: Please add horizontal and particularly, vertical scale bars.

An approximate outcrop size is not enough. Indicate the location of these outcrops on Figure 4. Add 'southeastward view of the Overgangshytta Fault' for the description of A in Figure caption. Location of 10D is not shown in 10A.

Comment 81: Figure 11: Please indicate profile location in Figure 4 and add approximate horizontal and vertical scales. Profiles like this greatly help the reader to follow the description of your observations and interpretations. It might be useful to refer to this figure earlier in the manuscript, e.g. in the results section.

Comment 82: Figure captions: - Replace crosscut with cross-cut where applicable

2. Author's response Comment 1: structural element and plate-tectonic reconstruction maps are probably not appropriate in such a short study with a relatively small study area. However, we believe that the comment of the reviewer is highly relevant to the next publication the main author is currently writing, which deals with the regional geology of Spitsbergen in the Mississippian and regional Cenozoic reactivation of Mississippian faults. In the study area, structural correlations are made over a maximum distance of 1 km in the field (Figure 4), 10–12 km for satellite images (Figure 1 and 3), and up to ca. 1000 km in the discussion when the findings of the present study is compared to recent findings in the NW Barents Sea (Anell et al., 2016) and in the SW Barents Sea (Koehl et al., 2018a). Comment 2: agreed. The size of scales and outcrops were added in figure captions were missing. However, the short duration of the fieldwork period in the area, and the number and quality of accessible outcrops did not always allow for quantitative measurements (only a few fault surfaces accessible for measurement; see stereonets in fig. 4). Comment 3: agreed. The present manuscript represents a relatively local study with greater implications than simply the geology of central Spitsbergen. However, the authors are aware of existing models (Braathen et al., 2011; Smyrak-Sikora et al., submitted) conflicting with their interpretation and would rather not extrapolate the results of such a small study area to the whole margin. Multiple disagreement in interpretation with initial co-authors of the manuscript (notably Prof. Olaussen, Dr. Smyrak-Sikora, and Dr. Johannessen – University Centre

in Svalbard – and Prof. Stemmerik – Natural History Museum Copenhagen) incites the authors of the present manuscript to cautiousness. Nevertheless, the main author is currently writing another manuscript focused on the regional geology of Spitsbergen in the Mississippian, using the findings of the present manuscript as supporting evidence to further argue for a regional model for the northern Barents Sea and west Spitsbergen margins. Regarding the use of field examples shown in the present study as analogues to subsurface studies in the Barents Sea, it is partly addressed in chapter 5.2, sub-chapter 3, paragraphs 3–5, in which reference to offshore studies is made (e.g., Anell et al., 2016; Phillips et al., 2016; Fazlikhani et al., 2017; Koehl et al., 2018a). Paragraph 3 compares an offshore study of the Gullfaks–Visund Fault (Cowie et al., 2005) to the Billefjorden Fault Zone, while paragraph 4 insists on the importance of Mississippian growth strata onshore Spitsbergen for seismic studies in the Barents Sea, notably building on the results of Anell et al. (2016) in the northwestern Barents Sea and their interpretation of thickened strata between basement and Permian strata. Paragraph 5 further compares offshore studies in Lofoten–Vesterålen (Bergh et al., 2007) and western Troms (Indrevær et al., 2013) to infer the extension direction.

Comment 4: agreed. However, the brief introduction of the succession of tectonic events at the beginning of the abstract is crucial for the reader to grasp the ambiguity of the scientific problem dealt with in the present manuscript (tectonic setting during the deposition of sedimentary rocks of the Billefjorden Group). Regional implications are not directly relevant to the present manuscript, although mentioned in the introduction chapter as suggested by the reviewer in subsequent comments, and will be dealt with in three upcoming manuscripts investigating contractional structures in sedimentary rocks of the Billefjorden Group in adjacent areas in central Spitsbergen (Koehl, in prep. b), and regional oblique-slip margin-oblique faults throughout Spitsbergen (Koehl et al., in prep) and Bjørnøya (Koehl, in prep. a).

Comment 5: the term "NNE-dipping" gives both the dip (to the NNE) and implies the strike (WNW–ESE) of the fault(s). This type of writing aims at keeping the manuscript

relatively short (although it is already long for the type of study and size of the study area). We hope it is alright to keep it this way throughout the whole manuscript.

Comment 6: agreed.

Comment 7: agreed.

Comment 8: agreed.

Comment 9: shale décollements decoupled deformation between lower basement faults and Pennsylvanian (to Cenozoic) sedimentary cover, and, thus, prevented further vertical movement along basement-seated faults.

Comment 10: agreed.

Comment 11: agreed.

Comment 12: agreed.

Comment 13: agreed. However, the use of a regional cross-section might not be this useful for such a local study. Nevertheless, the first author of the present manuscript is currently writing another manuscript on the same topic at a regional scale in Spitsbergen and will use the suggestion of the Dr. Lenhart in this future manuscript.

Comment 14: disagreed. Again, this manuscript is a very local study and crowding an already quite long manuscript with regional maps an tectonic reconstructions might not be appropriate, but it may be relevant for the first author's upcoming regional manuscript.

Comment 15: agreed.

Comment 16: agreed. Very good point, the manuscript is not clear enough.

Comment 17: agreed.

Comment 18: agreed.

Comment 19: agreed.

Comment 20: disagreed. The current sentence illustrates better our point in that the satellite images where carefully selected because of the relevance of the area they cover, not because the area was difficult to access.

Comment 21: agreed. However, the faults crosscutting the Atomfjella Antiform mentioned in this sentence are located north of the area shown in figure 1b (see Witt-Nilsson et al., 1998) and can therefore not be included on the map.

Comment 22: agreed for Ny-Friesland and the Atomfjella Antiform (now shown in figure 1a and 1b respectively). However, smaller localities like Mittag-Lefflerbreen are already mentioned in figure 3 and would rather overcrowd figure 1.

Comment 23: agreed.

Comment 24: agreed.

Comment 25: agreed.

Comment 26: agreed.

Comment 27: disagreed. "On the contrary" better illustrate our point.

Comment 28: agreed.

Comment 29: agreed.

Comment 30: agreed.

Comment 31: agreed.

Comment 32: agreed.

Comment 33: agreed.

Comment 34: agreed.

Comment 35: agreed.

Comment 36: agreed. However, this topic cannot be addressed in the result chapter and the

comment was implemented in the first subchapter of the discussion.

Comment 37: agreed. However, as mentioned in our response to previous comments, the present manuscript is a local study that will represent the corner stone of a regional study in Spitsbergen. The wider implications will be addressed in this next manuscript.

Comment 38: agreed.

Comment 39: agreed. Highly relevant comment, which led to a reorganization of sub-chapter 5.1 and to a significant improvement of the manuscript.

Comment 40: agreed.

Comment 41: agreed. However, the denomination "down-NNE" is often used in similar scientific articles and the authors would therefore prefer to keep the formulation this way.

Comment 42: agreed. The manuscript currently lacks reference to relevant paleo-tectonic reconstructions. However, the authors would prefer not to include any plate tectonic reconstruction map to the manuscript because it is nor the aim neither part of the results of the manuscript.

Comment 43: agreed.

Comment 44: agreed.

Comment 45: agreed.

Comment 46: agreed.

Comment 47: this topic is addressed in paragraph number 5 of the last sub-chapter of the discussion ("Switch from widespread to localized extension").

Comment 48: agreed.

Comment 49: agreed.

Comment 50: agreed.

Comment 51: agreed.

Comment 52: agreed.

Comment 53: comment addressed in our response to comment 42.

Comment 54: comment addressed in our response to comment 47.

Comment 55: agreed.

Comment 56: agreed.

Comment 57: the authors used "(a)" and "(s)" to show that the observed tilting might results from displacement along one or several faults. However, this formulation does not seem to be clear enough and the authors addressed the issue.

Comment 58: agreed.

Comment 59: agreed.

Comment 60: agreed.

Comment 61: the Finnmark Platform is located some 800 km away from the study area, i.e., the study area and the Finnmark Platform and closer to each other than the Caledonides of northern Norway and the Caledonides of Svalbard. Although our correlation might seem farfetched right now, the correlation of the Caledonides across the North Atlantic Ocean and the Barents Sea might have been farfetched too a few decades ago. Moreover, multiple studies tend to suggest such Timanian affinity is possible (see Mazur et al., 2009; Majka et al., 2010; Klitzke et al., 2018, submitted; Koehl, in prep.).

Comment 62: the authors believe that the extension direction was constant (see Bergh et al., 2007; Eig and Bergh, 2011; Hansen and Bergh, 2012; Koehl et al., 2018) and, alone, may explain all the observed fault patterns and kinematics.

Comment 63: agreed. However, the thickness of the coaly beds in the Billefjorden Group is already extensively mentioned in the result chapter, section 4.2, paragraph 1.

Comment 64: agreed.

Comment 65: agreed.

Comment 66: agreed.

Comment 67: disagreed. Again, the present manuscript is a local study with regional implications. However, the regional implications would be too farfetched if the authors were to propose a regional model for Spitsbergen and the Barents Sea only based on a local field and remote sensing study. Regarding the "complexity" of the conclusion points, these will be the foundations of two upcoming manuscript and, thus, need to be very specific and detailed in order for the reader to link the present manuscript to upcoming work.

Comment 68: agreed.

Comment 69: agreed.

Comment 70: agreed.

Comment 71: agreed. However, the present manuscript is a local study targeting a small audience of (geo-) scientists working with Svalbard and the Arctic. Thus, the authors argue that a regional map with structural lineaments may not be appropriate to include. Such maps may be found in Bergh et al. (2007), Indrevær et al. (2013), Anell et al. (2016), Koehl (2018) and Koehl et al. (2018a, 2018b).

Comment 72: agreed.

Comment 73: agreed. However, it is not possible to improve the quality of the satellite images.

Comment 74: yes, the dip of some of the faults interpreted from the satellite images is unknown. Pink and blue double-arrows indicate outcrop exposures of the Hultberget Formation and Billefjorden Group respectively, as indicated in the caption of figure 4.

Comment 75: disagreed. The person in the lower right corner is the scale. In addition, the label "figure 5b" in figure 5a correctly indicates the location of figure 5b.

Comment 76: agreed. However, vertical and horizontal scale being the same, there is no need to add both.

Comment 77: agreed. However, vertical and horizontal scale being the same, there is no need to add both.

Comment 78: agreed.

Comment 79: agreed.

Comment 80: agreed.

Comment 81: agreed. However, figure 11 is a schematic N–S profile across the study area shown in figure 4. Adding a line to show the approximate location of the profile would crowd figure 4 too much. Figure 11 is the proposed model for the study area and is quite interpretative and sometimes speculative. Thus, it might not be judicious to mention it in the result chapter.

Comment 82: agreed.

3. Changes implemented Comment 1: added references to figures throughout the main text.

Comment 2: added "Person as scale in the lower right corner" in the caption of fig. 2a; "Rifle orange cover as scale (ca. 1.20 m-long)" in the caption of fig. 2c; "Camera cover

(15x10 cm) as scale" in the caption of fig. 2d; "and 2–2.5 m high" in the caption of fig. 6; "The outcrop is approximately 10 m high" in the caption of fig. 7a; "The outcrop is ca. two meters high" in the caption of fig. 7b; "The outcrop is ca. three meters high" in the caption of fig. 7c; "shows the width of the core" in the caption of fig. 10a; "The fault core is limited by the dashed white and dashed red lines and is ca. 3 meters wide" in the caption of fig. 10e; "Ca. one km-long" in the caption of fig. 11.

Comment 3: no change.

Comment 4: shortening of the last two sentences of the abstract: deletion of ", thus suggesting that normal faulting along this major fault initiated as early as the Mississippian" lines 36–37, and of "Mississippian margin-oblique" line 40.

Comment 5: no change.

Comment 6: implemented suggested change.

Comment 7: implemented suggested change.

Comment 8: implemented suggested changes.

Comment 9: no change.

Comment 10: addition of a few lines on regional implications lines 60–70.

Comment 11: implemented suggested change.

Comment 12: addition of the Atomfjella Antiform, Odellfjellet Fault, Balliolbreen Fault, and Løvehovden Fault to figure 1b.

Comment 13: addition of a few key structural elements to figure 1b (see response to comment 12), and addition of all the outcrop photograph location on figure 4.

Comment 14: no change.

Comment 15: implemented suggested change.

Comment 16: addition of "west-directed thrusting" lines 150–151.

Comment 17: implemented suggested change.

Comment 18: implemented suggested change.

Comment 19: addition of 10 lines (lines 215–224) on the satellite photograph resolution and on the interpretation methodology with regards to field outcrops.

Comment 20: no change.

Comment 21: addition of the location of "Ny-Friesland" in figure 1a

Comment 22: Ny-Friesland and the Atomfjella Antiform are now shown in figure 1a and 1b respectively.

Comment 23: implemented suggested change.

Comment 24: added thickness of beds.

Comment 25: implemented suggested change.

Comment 26: implemented suggested change.

Comment 27: no change.

Comment 28: addition of a scale in figure 7 and of the bed thickness in the relevant paragraph.

Comment 29: addition of "and offsets are generally decimeter- to meter-scale (Figure 8)" line 307.

Comment 30: addition of "tens-of-centimeter-thick" lines 311–312.

Comment 31: implemented suggested change.

Comment 32: implemented suggested change.

Comment 33: implemented suggested change.

Comment 34: implemented suggested change.

Comment 35: implemented suggested change.

Comment 36: addition of "comprised between a few meters and" line 391.

Comment 37: no change.

Comment 38: deletion of "made of sedimentary strata of the Hultberget, Ebbadalen and Minkinfjellet formations" lines 359–360, and changed "thus suggesting" into "which suggests" line 361.

Comment 39: The third paragraph of sub-chapter 5.1 was moved to the beginning of the sub-chapter. The authors also added reference to quantitative studies to the main text lines 371–375 "This is supported by quantitative studies on the width of fault cores (e.g., Forslund and Gudmundsson, 1992; Childs et al., 2009; Bastesen and Braathen, 2010; Johannessen, 2017), which indicate that faults with 2–3 meters wide core zones (like the Overgangshytta fault; Figure 10a) generally accommodate vertical displacement ranging from a few meters to several hundreds of meters", lines 383–386 "Notably, quantitative studies discussing potential relationships between fault length and displacement show that a fault like the Overgangshytta fault is likely to be several hundred to a few thousand meters long (Watterson, 1986; Nicol et al., 1995; Schlische et al., 1996; Gudmundsson, 2000; Kolyukhin and Torabi, 2012)", and to the reference list.

Comment 40: implemented suggested change.

Comment 41: changed "km-thick" into "kilometer-thick" line 360.

Comment 42: addition of "Although not always reconstructed in paleo-tectonic reconstructions, in the early Neoproterozoic, the position of Svalbard was probably close to the Timanian margin of northern Baltica prior to the opening of the Asgard Sea and Iapetus Ocean/Ægir Sea (Torsvik et al., 1996; Cawood et al., 2001, 2010; Cawood and Pisarevsky, 2017), and prior to the Timanian Orogeny in the late Neoproterozoic

(Roberts and Siedlecka, 2002; Roberts and Olovyanishnikov, 2004)" lines 420–426 and to the reference list. In addition, the authors added a sentence about the Timanian Orogeny in the introduction lines 84–87.

Comment 43: addition of ", potentially accommodating a few meters to several tens of meters of reverse displacement" lines 482–483.

Comment 44: implemented suggested change.

Comment 45: replacement "small" by "meter" line 484.

Comment 46: addition of "(centimeter- to decimeter-scale)" line 484.

Comment 47: no change.

Comment 48: addition of "thickened by several tens of centimeters" line 514.

Comment 49: implemented suggested change.

Comment 50: implemented suggested change.

Comment 51: implemented suggested change.

Comment 52: implemented suggested change.

Comment 53: see response to comment 42.

Comment 54: see response to comment 47.

Comment 55: implemented suggested change.

Comment 56: addition of "gentle (10–30°)" to the result chapter line 263 and to the discussion chapter lines 580–581.

Comment 57: replacement of "(a)" line 591 (two occurrences) by "one or several". Deletion of "(s)" lines 591 and 592.

Comment 58: addition of "B" and "K" in figure 1a to locate Brøggerhalvøya (B) and

Kongfjorden (K).

Comment 59: implemented suggested change.

Comment 60: addition of "ENE–WSW-oriented" line 203 in the geological setting chapter and line 617 in the discussion chapter, and of "west-directed" and "thrusting" in the discussion chapter lines 616 and 617.

Comment 61: no change.

Comment 62: no change.

Comment 63: added the suggested reference to the reference list and to the main text lines 488–497.

Comment 64: implemented suggested change.

Comment 65: addition of "(< 1 km)" line 664.

Comment 66: implemented suggested change.

Comment 67: no change.

Comment 68: implemented suggested changes.

Comment 69: implemented suggested change.

Comment 70: implemented suggested change.

Comment 71: addition of multiple localities to figures 1a and 1b.

Comment 72: implemented suggested change.

Comment 73: implemented suggested change.

Comment 74: no change.

Comment 75: no change.

Comment 76: implemented suggested change.

Comment 77: implemented suggested changes.

Comment 78: implemented suggested changes.

Comment 79: implemented suggested changes.

Comment 80: replacement of "Outcrop photograph showing the geometry" by "Eastward view" line 1231.

Comment 81: addition of a scale bar to figure 11.

Comment 82: implemented suggested changes.